# The role of promiscuous molecular recognition in the evolution of RNase-based self-incompatibility in plants

Keren Erez[1,3], Amit Jangid [1,3], Ohad Noy Feldheim [2] & Tamar Friedlander [1] ✉

How do biological networks evolve and expand? We study these questions in the context of the plant collaborative-non-self recognition self-incompatibility system. Self-incompatibility evolved to avoid self-fertilization among hermaphroditic plants. It relies on specific molecular recognition between highly diverse proteins of two families: female and male determinants, such that the combination of genes an individual possesses determines its mating partners. Though highly polymorphic, previous models struggled to pinpoint the evolutionary trajectories by which new specificities evolved. Here, we construct a novel theoretical framework, that crucially affords interaction promiscuity and multiple distinct partners per protein, as is seen in empirical findings disregarded by previous models. We demonstrate spontaneous self-organization of the population into distinct "classes" with full between-class compatibility and a dynamic long-term balance between class emergence and decay. Our work highlights the importance of molecular recognition promiscuity to network evolvability. Promiscuity was found in additional systems suggesting that our framework could be more broadly applicable.

Hermaphroditic flowering plants are at high risk of self-fertilization, which would produce less fit offspring (known as "inbreeding depression"[1]). Hence, more than 100 flowering plant families have developed various mechanisms to avoid self-fertilization, generally called "self-incompatibility" (SI)[2–10]. Under these mechanisms, the species is sub-divided into multiple "types" or "classes", such that a pollen grain cannot fertilize a maternal plant of its own type. The type is encoded by a single highly polymorphic locus called the S-locus that encodes both male (Pollen-S) and female (Pistil-S) type-specifying genes.

The molecular mechanisms implementing type recognition are categorized as using either "self" or "non-self recognition"[6,7,9,11]. Under self-recognition (SR), fertilization is by default enabled unless a maternal plant identifies an incoming pollen as having the same type as its own. Hence this mechanism requires only a single type-identifier. In contrast, under non-self recognition (NSR), fertilization is by default disabled, and only if the incoming pollen is positively identified as having a non-self type, fertilization is unlocked. Thus, this mechanism requires multiple identifiers, that could collectively identify multiple non-self types.

Here we focus on the collaborative non-self recognition (CNSR) SI mechanism demonstrated in the Solanaceae family (tomato, potato, tobacco, *Petunia*) and on homologous mechanisms found in Maloideae of Rosaceae (apple, pear, loquat) and Plantaginaceae (snapdragon) families, also known as the "RNase-based SI". The female-determinant in this mechanism is a cytotoxic S-locus encoded ribonuclease (S-RNase)[12]. S-RNase molecules expressed in female organs are imported into growing pollen tubes that attempt to fertilize the maternal plant. If the pollen is compatible, the S-RNase molecules are recognized by its male-determinant proteins and degraded, allowing for fertilization. Otherwise, if the pollen is incompatible, the S-RNases arrest the pollen tube growth, and fertilization is inhibited. The male-

[1]The Robert H. Smith Institute of Plant Sciences and Genetics in Agriculture, Faculty of Agriculture, The Hebrew University of Jerusalem, P.O. Box 12, Rehovot 7610001, Israel. [2]The Einstein Institute of Mathematics, Faculty of Natural Sciences, The Hebrew University of Jerusalem, Jerusalem 9190401, Israel. [3]These authors contributed equally: Keren Erez, Amit Jangid. ✉e-mail: tamar.friedlander@mail.huji.ac.il

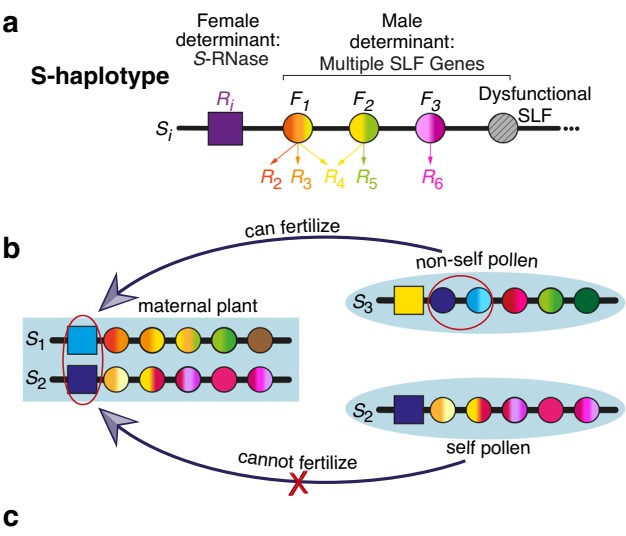

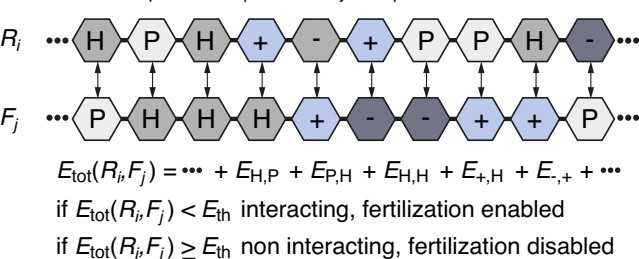

**Fig. 1 | Self-incompatibility in the S-RNase-based mechanism – the collaborative non-self recognition (CNSR) mechanism. a** The S-locus includes a single S-RNase gene (expressed in the female organs) and multiple SLF paralogs (expressed in the male organs). An SLF protein can detoxify one or more different S-RNase proteins, but could also detoxify none, in which case it is considered dysfunctional. **b** A haploid pollen harboring a particular combination of SLF proteins can successfully fertilize a diploid maternal plant only if it is equipped with the specific SLFs that can detoxify the maternal plant's two S-RNases (top, encircled SLFs). An S-locus usually does not contain SLF genes capable of detoxifying its own S-RNase, hence its pollen cannot fertilize its own ovules. Such a haplotype is called "self-incompatible" (bottom), whereas a haplotype that does contain SLF compatible with its own S-RNase is called "self-compatible". Due to inbreeding depression, there is a strong selection pressure against self-compatibility[10]. **c** The protein–protein interaction model: we assume that the total interaction energy between two proteins $R_i$ and $F_j$ is the sum of the pairwise interaction energies between their corresponding amino acids. A pair of RNase and SLF proteins are considered interacting (fertilization is enabled) only if this energy is below a threshold value $E_{th}$. Otherwise, if $E \geq E_{th}$ they are considered non-interacting, and fertilization is disabled.

determinant in this mechanism has been identified as an F-box protein-encoding gene and was termed S-locus F-box (SLF or SFB or SFBB)[13,14]. Relying on NSR, indeed multiple SLF genes are encoded in the S-locus to collaboratively recognize various non-self S-RNases and allow for a sufficient number of mating partners[11]. To avoid self-fertilization, a haplotype must not contain an SLF variant that recognizes its own (self) RNase, which would lead to self-compatibility. Indeed, the S-locus genes are tightly linked to avoid an accidental insertion of an SLF that could cause self-compatibility, as well as preserve the collaborative function of the entire haplotype. In the following, we refer to this mechanism as the "collaborative non-self recognition" (CNSR) SI (Fig. 1).

Clearly, this system requires molecular recognition that could distinguish between different molecular types, but how exactly the specificity of different RNase and SLF proteins is encoded remains elusive[15]. A combination of structural modeling[16], targeted mutations[17–19], domain swapping experiments[19,20], and selection footprints analysis[21–24] revealed that very few positions determine these protein specificities.

Despite the progress in understanding the molecular mechanisms underlying SI, a conceptual puzzle remains: while empirical evidence shows high allelic diversity, it remains murky how novel specificities evolve and which factors determine their numbers in natural populations. Early works assumed that a single mutation is sufficient to produce a new specificity[3,25]. In these models, the equilibrium number of distinct specificities ("S-alleles") was determined by the balance between their introduction via new mutations and their loss by drift. Discovery of the S-locus genetic architecture led later modelers to assert that a combination of at least two mutations – in both female and male determinants – was needed to produce a new specificity. To address the conundrum of how this combination of mutations could survive, it was proposed that a new specificity could emerge via a self-compatible intermediate which later mutated to restore SI[26–28]. Nevertheless, all the aforementioned models assumed SR-based SI, in which a newly emerging specificity was always compatible with the existing ones and hence selectively advantageous owing to its rarity.

The case for CNSR is different because a new specificity is not necessarily compatible with existing ones. Thus, the process allowing new specificities to establish in the population, once they emerge, appears even more mysterious. The hurdle is that individuals carrying a novel S-RNase allele having no compatible SLF in the population are sterile as females, while novel SLFs that have no matching RNase yet are futile and hence their expansion in the population is only neutral. Moreover, due to the tight linkage between the S-locus genes, it is insufficient that the compatible SLF appears just once, but should rather appear multiple times on all genetic backgrounds, but the one carrying the new RNase. An earlier emergence of a compatible SLF could enable the invasion of the new RNase. Yet, in the likely occasion that this compatible SLF has only spread to part of the population, the remaining part incompatible with the new RNase is prone to extinction following the RNase invasion[29,30]. Thus, the overall outcome of the emergence of a new RNase is not necessarily an augmentation of the total number of specificities but may just as well be its reduction.

These considerations place severe constraints on possible models for the evolution of new specificities in the CNSR system. Previous theoretical models that focused on the unique challenges of CNSR evolution either suggested that the novel SLF appeared multiple times on different genetic backgrounds via repeated mutations[31], or that the compatible SLF appeared just once and then spread horizontally to additional genetic backgrounds via gene conversion[7,29]. Both models considered the scenario that the novel RNase appeared first to be deleterious and proposed that the novel SLF should precede the RNase.

Previous models simplistically assumed one-to-one interactions between particular RNases and SLFs. While this is indeed the case for SR, recent empirical evidence suggested a more complicated picture for CNSR, where a particular SLF could potentially recognize several distinct RNases with possible overlaps in the recognition capacities of different SLFs[7,11,21,32–35].

Richer biophysical models accounting for between-residue interactions were used to study protein folding (known as lattice models)[36–38], the evolution of protein–protein interaction networks[39–41], molecular recognition in receptor repertoires[42], and immune recognition[43–45]. To account for between-residue interaction energies, these models used either the Miyazawa–Jernigan potentials[39,45,46] or coarse-grained descriptions with only two or four amino acid categories[37,38,40,43]. Similar biophysical models for protein–DNA interactions were incorporated into evolutionary models to study the evolution of gene regulation[47,48].

Building on these evolutionary-biophysical models, we formulate a framework, which accounts for the energetic interactions between

RNase and SLF proteins, and allows mating between individuals based on matches between their protein content. This framework offers several unique properties leading to surprising outcomes that were not possible before. First, multiple genotypes could map into a common compatibility phenotype. Hence, a large proportion of mutations are neutral, and in particular neutral RNase mutations become possible. Second, our model defines a "compatibility landscape", on which genotypes that share a phenotype are connected via neutral networks and the compatibility phenotypes of neighboring genotypes are correlated, reminiscent of RNA secondary structures[49]. Third, the model assumes significant promiscuity of interactions between proteins. These properties crucially allow for many-to-many interactions between RNase and SLF proteins, in agreement with empirical evidence.

Most haplotypes in our model spontaneously self-organize into "compatibility classes", such that members of each class are incompatible with each other but compatible with all members of all other classes. Such classes are regularly born and die. Essentially neutral RNase mutations not only occur regularly but turn out to be mandatory for the dominant class emergence trajectories. Owing to promiscuity, the model exhibits a dynamic balance between class birth and death and shows a stable equilibrium in their number. These behaviors prevail under a broad range of parameters. We propose that these are key features of the natural system, and that such evolutionary trajectories may offer a solution to the conundrum of the evolution of new SI specificities in the CNSR system.

Below, we describe our model in detail, show the various trajectories for class birth and death, and demonstrate their dynamics, as found in simulations.

## Results

### An evolutionary-biophysical model for the formation and extinction of compatibility classes

We consider a population of $N$ diploid individuals, each composed of two haplotypes. Every haplotype contains a single RNase encoding the female-specificity, and multiple SLF paralogs, encoding the male-specificity (Fig. 1a). Every diploid individual plays the role of a diploid maternal plant, as well as produces two types of haploid pollen carrying either of its two haplotypes. A haploid pollen can successfully fertilize a diploid maternal plant only if it is equipped with the appropriate SLFs that can successfully detoxify the maternal plant two RNases (Fig. 1b). It is then considered "compatible" as a sire with both maternal haplotypes. Compatibility between two haplotypes could, in general, differ from their compatibility when their sire-dam roles are switched. We distinguish between unidirectional and bidirectional (in) compatibility, where the latter means the two haplotypes are (in) compatible in both roles.

For simplicity, we construct the model directly in the protein domain. We represent every RNase or SLF by a sequence of $L$ amino acids, standing for the binding domain of that protein. We use a size-4 alphabet representing four biochemical classes of amino acids: hydrophobic (H), neutral polar (P), positively charged (+), and negatively charged (−). We define the total interaction energy between an RNase $R_i$ and an SLF $F_j$ as the sum of pairwise interaction energies between their corresponding residues $E_{tot}(R_i,F_j) = \sum_{k=1}^{L} E(R_i(k)F_j(k))$ using values from ref. 38 (Table 2). These proteins are then considered matching if the total interaction energy is below a preset threshold value (Fig. 1)

$$E_{tot}(R_i,F_j) < E_{th}. \qquad (1)$$

We tested several values of $E_{th}$ (Figs. S4, S5) and $L$ (not shown) and verified that the qualitative behavior of the model is persistent. For most of the results described below, these parameters were chosen following interaction energy distributions as in ref. 50.

By construction, this model enables distinct sequences to have different numbers of matching partners, or possibly not to have such partners at all. Note also the intricate role of mutations in this model, where not every mutation essentially alters compatibility.

We simulated the evolution of a population of $N$ such diploid individuals with the following life cycle (Fig. 2). Every generation, each of the SLFs (but not the RNase) could be duplicated with probability $p_{dup}$ and deleted with probability $p_{del}$, allowing for variation in the number of SLF paralogs per haplotype (Fig. S20). Each residue in any of the RNases or SLFs can be mutated with probability $p_{mut}$.

A haploid pollen can successfully fertilize a diploid maternal plant only if its SLFs can collaboratively detoxify both maternal plant RNases, following Eq. (1). To produce the population next generation we repeatedly pick a maternal plant uniformly at random (with replacement). If both maternal plant haplotypes are self-incompatible, we grant it multiple opportunities to mate with uniformly chosen non-self pollen grains. This sire-dam asymmetry represents the typically higher abundance of pollen grains relative to ovules. The first compatible pollen produces one offspring with one of the maternal plant two haplotypes. If any of the maternal plant haplotypes are self-compatible, it can also be self-fertilized with a certain probability, but the resulting offspring survives with a lesser probability (relative to outcrossing offspring), due to inbreeding depression (Methods). We repeat this procedure until $N$ offspring are formed. Finally, the offspring population replaces the parental population, completing one generation. See list of default parameter values in Table 1.

### Dynamic balance between emergence and extinction of genetically heterogeneous compatibility classes

We ran the stochastic simulation multiple times for 100,000–150,000 generations each. The analyses below include only times beyond the point that the entire population descended from a single ancestral haplotype, in which its distribution is nearly independent of the initial conditions used.

We classified the vast majority of the $2N$ population haplotypes into compatibility classes, defined such that every pair of classified haplotypes were bidirectionally incompatible if they were in the same class, and bidirectionally compatible if in different classes. Importantly, classes are defined only based on the compatibility phenotype and hence class members could be genetically heterogeneous (Fig. 3).

While class definition does not necessitate that every haplotype belongs to a class, and some may be left out, we found that on average only 5% of the haplotypes remained unclassified, and only in 9% of the instances <90% were classified (Fig. S3). We associated each unclassified haplotype with an "appendix" of the class it most recently belonged to. This small fraction of unclassified haplotypes often remained unclassified for only short time periods, typically shifting back and forth between the class and its appendix. Nearly all unclassified haplotypes either lacked one interaction with one of the foreign classes (previously termed "incomplete"[31]) or had one excess interaction with some of their former class members (Methods). The number of distinct female specificities should be at least as large as the number of classes. In the following, we focus on the number of population classes, as a proxy for the number of specificities.

Next, we asked whether the number of population classes reached a stable equilibrium value with possible fluctuations around it or alternatively oscillated between high and low values. To address this question, we analyzed the population class structure every 10 generations and calculated the distribution of the number of classes amongst different population instances (Fig. 4a). We found that this distribution is unimodal and stable in time and across different runs, and hence described the process' stationary distribution.

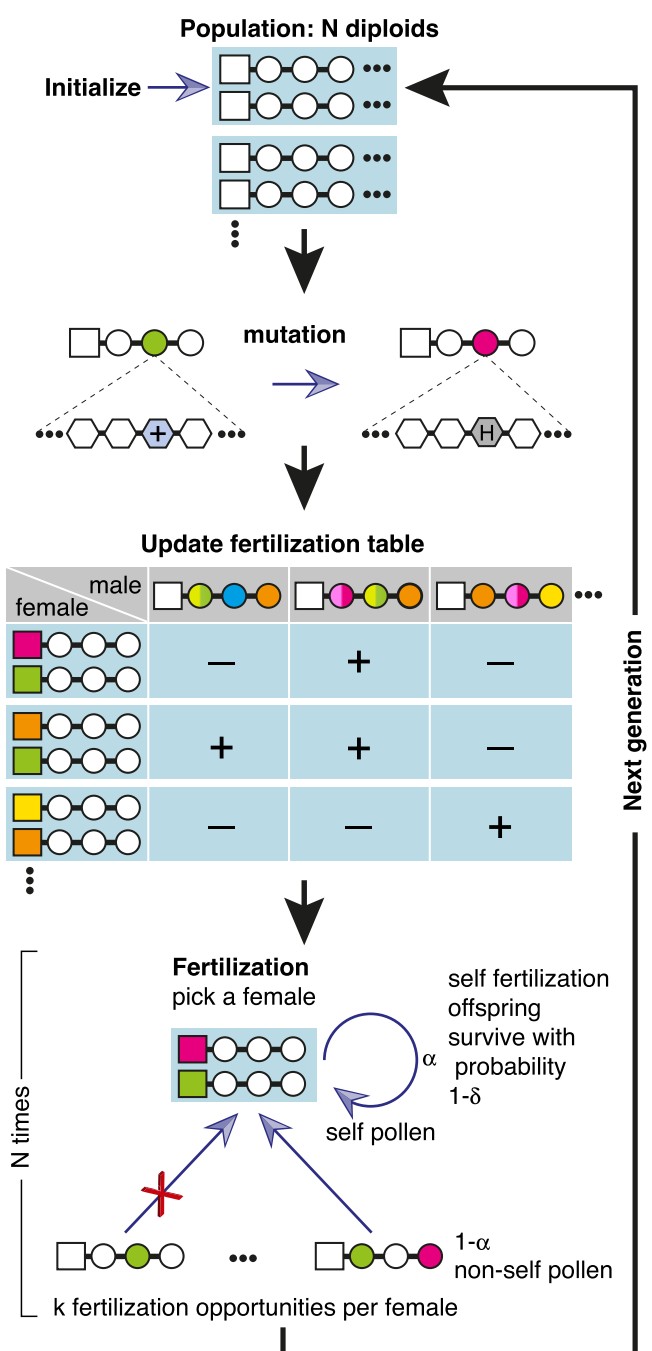

**Fig. 2 | The population life cycle in our simulation.** We initialize a population of $N$ self-incompatible diploid heterozygous individuals, where each haplotype is composed of a single RNase (square) and multiple SLF genes (circles), each represented by a sequence of $L$ amino acids (Fig. 1). Every generation, each of these genes could be mutated via the substitution of randomly chosen residues. A diploid maternal plant can be fertilized by a haploid pollen, only if the pollen is equipped with suitable SLFs that can detoxify the two maternal S-RNases. To keep track of all compatible pairs, we chart the table of possible crosses between all haploid pollen and diploid maternal plant combinations. We assume that a proportion $\alpha$ of the pollen received by each maternal plant is self-pollen and the remaining $1-\alpha$ proportion is foreign pollen. If an individual is self-compatible, only a proportion $1-\delta$ of the offspring produced by self-fertilization survives. We then draw the next generation of the population by randomly picking maternal plants (with replacement) and then granting each $k$ opportunities to match a randomly chosen pollen. If a matching pollen is found within $k$ attempts, the maternal plant and the first successful pollen produce one offspring. This process is continued until a population of $N$ offspring is formed, which then replaces the parental population. Each cycle represents a single generation. The default parameter values are: population size $N = 500$, per-residue mutation rate $p_{mut} = 10^{-4}$ per generation, number of generations 100,000–150,000, $\alpha = 0.95$, $\delta = 1$.

### The most prevalent split trajectories begin with a neutral RNase mutation followed by two SLF mutations

We further investigated the sequence of events leading to class birth and death. We found out that class births always occurred via the splitting of an existing class ("mother class") into exactly two classes. In the following we refer to class birth as "split" and to class death as "extinction". We identified three main routes for split and three for extinction that could sometimes intertwine, in all of which an RNase mutation played a pivotal role (Fig. 5a, b). The formation of a new class requires an RNase mutation in the mother class, followed by an SLF mutation on the background of the new RNase, which renders this haplotype compatible as sire with the original female-specificity of the mother class. To evade extinction, the latter must undergo an analogous SLF mutation rendering it compatible as sire with the novel class female-specificity. The three different trajectories described schematically in Fig. 5a correspond to the three different orderings in which the latter SLF mutation on the background of the original RNase can appear with respect to the former two mutations: after (first trajectory, blue), in-between (second trajectory, magenta), or before (third trajectory, yellow) – colors refer to the paths as illustrated in Fig. 5a. In all trajectories the first mutation is essentially neutral (Supplementary Note 1) and the second one confers a fitness advantage and could drive some haplotypes to extinction unless rescued by the third mutation. In principle, the three mutations needed for a split could occur in six different orderings, but the other three options pass through a self-compatible intermediate and hence are much less common under the high inbreeding depression needed to maintain SI. In Fig. 5c, we show the simulation frequencies of the six optional split trajectories. These could potentially vary depending on the simulation parameter values and specifically inbreeding depression (see Figs. S15–S17). The two most common trajectories start with the RNase mutation as the first and neutral one. Neutral RNase mutations are a unique feature of our model, made possible by the multi-specificity of proteins. Only the third trajectory, which starts with a neutral SLF mutation, was previously studied[31], whereas the first and second ones are described here for the first time.

In Fig. 6, we illustrate the split dynamics of the most prevalent split trajectory using an example from our simulations (see the other trajectories in Fig. S28). For every trajectory, we present three plots showing (from top to bottom) the copy numbers of the relevant haplotypes $X_i$, their male-fitness $f_i$, and "fitness-adjusted copy numbers" $\tilde{X}_i = X_i + (f_i - \bar{f}) \cdot N$ (Methods) against time, where $\bar{f}$ is the population mean male-fitness. Haplotype male-fitness $f_i$ is defined as the proportion of diploid individuals in the population the haplotype is compatible with as sire. While reproductive capacity is affected by

To investigate the class number dynamics we traced the time points at which it either increased or decreased and designated them as class "birth" and "death" events, respectively (Fig. 4b). Naturally, if transitions are only possible between adjacent class numbers, the upward and downward transitions between each pair of consecutive class numbers should exactly balance each other. We observed only minor deviations from such an exact balance. Indeed, classes are always born one at a time and simultaneous extinctions of multiple classes are possible but rare.

The class birth and death rates exhibit markedly different dependence on the current class number (Fig. 4c): below the most probable class number the birth rate supersedes the death rate, and the opposite happens above this number. Hence this reversal point is a unique stable equilibrium state (see Supplementary Text). This picture differs from the mass allelic extinctions, proposed in previous works[29]. We elaborate below on the reasons for this difference.

**Table 1 | Notation and default parameter values in the simulation**

| Parameter | Description |
|---|---|
| $N = 500$ | Population size |
| $n_h = 10$ | Initial number of haplotype types |
| $n_h - 1 = 9$ | Initial number of SLF paralogs per haplotype |
| $L = 18$ | Number of residues in each gene |
| $E_{th} = -6$ | Interaction energy threshold below which an RNase and SLF are considered interacting |
| $p_{dup} = 10^{-6}$ [1/generation] | Gene duplication rate |
| $p_{del} = 10^{-6}$ [1/generation] | Gene deletion rate |
| $p_{mut} = 10^{-4}$ [1/generation] | Per-residue mutation rate |
| $\alpha = 0.95$ | Proportion of self-pollen received by a diploid maternal plant |
| $\delta = 1$ | Proportion of offspring resulting from self-fertilization that do not survive |
| $k = 5$ | The number of fertilization attempts per diploid maternal plant |

Genetically homogeneous classes

Genetically heterogeneous classes

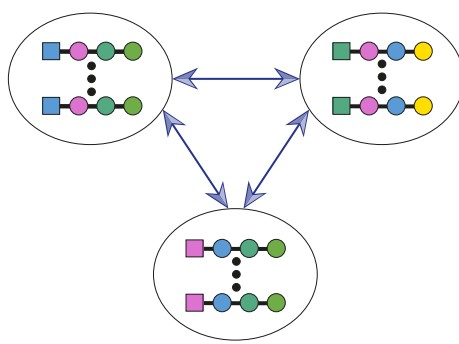 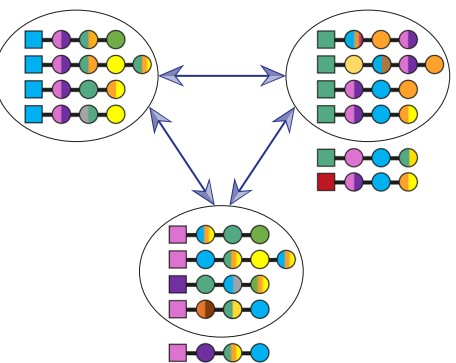

**Fig. 3 | Compatibility classes are genetically homogeneous under one-to-one interactions, but could be genetically heterogeneous under the many-to-many interaction model.** Compatibility classes are defined such that all members of each class are bidirectionally incompatible within the class, but simultaneously bidirectionally compatible with all members of all other classes. Previous models assuming that interactions between RNase and SLF are one-to-one, implied that compatibility classes should be genetically homogeneous, except for useless SLFs (left). Here, in contrast, we incorporate a more intricate interaction model, in which each protein could potentially have multiple different matching partners. Hence, within-class genetic heterogeneity becomes possible (right). Crucially, haplotypes in the same class could differ not only in their SLFs, but also in their RNase alleles. Following this definition, some haplotypes might remain unclassified. We illustrate a few examples of unclassified haplotypes (external to the ellipses): an RNase (red) that does not have a matching SLF in any of the classes, a self-compatible haplotype (green RNase and green SLF), and a haplotype bearing an SLF that matches other class members (purple SLF).

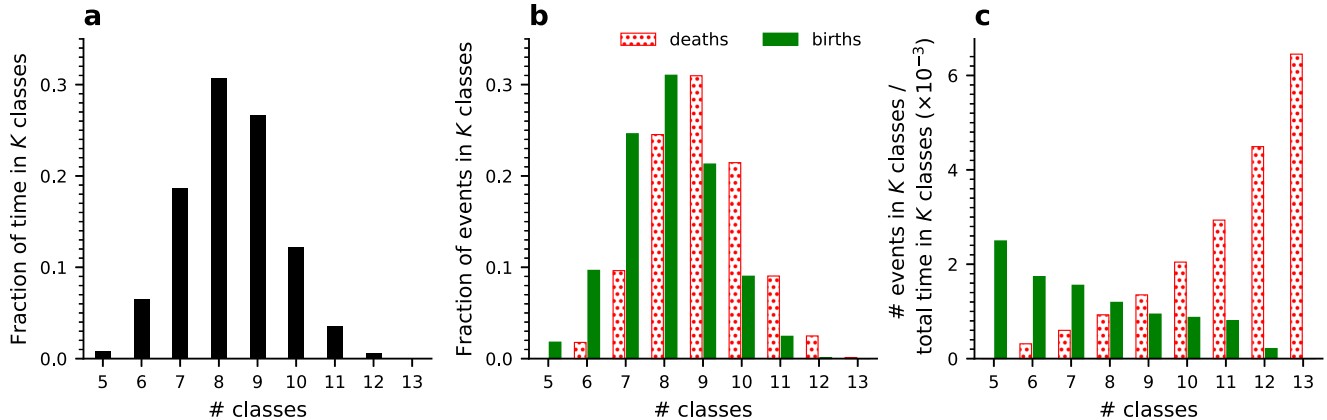

**Fig. 4 | Dependence of class birth and death events on the present number of classes suggests a stable equilibrium in the number of classes – simulation results. a** The fraction of time spent under each $K$-class population state. **b** The fraction of birth (green) and death (red) events that occurred under each $K$-class population state. **c** The event rates: the number of class birth (green) and class death (red) events that occurred under each $K$-class state divided by the time spent in this state. We observe opposite trends of these rates such that the class birth rate decreases but the class death rate increases with extant class number, suggesting a stable equilibrium at an intermediate value. The proportion of unclassified haplotypes and the proportion of self-compatible ones are shown in Fig. S3. For simulation parameters see Table 1. Results are based on 39 independent runs, with a total of 1,478,050 generations.

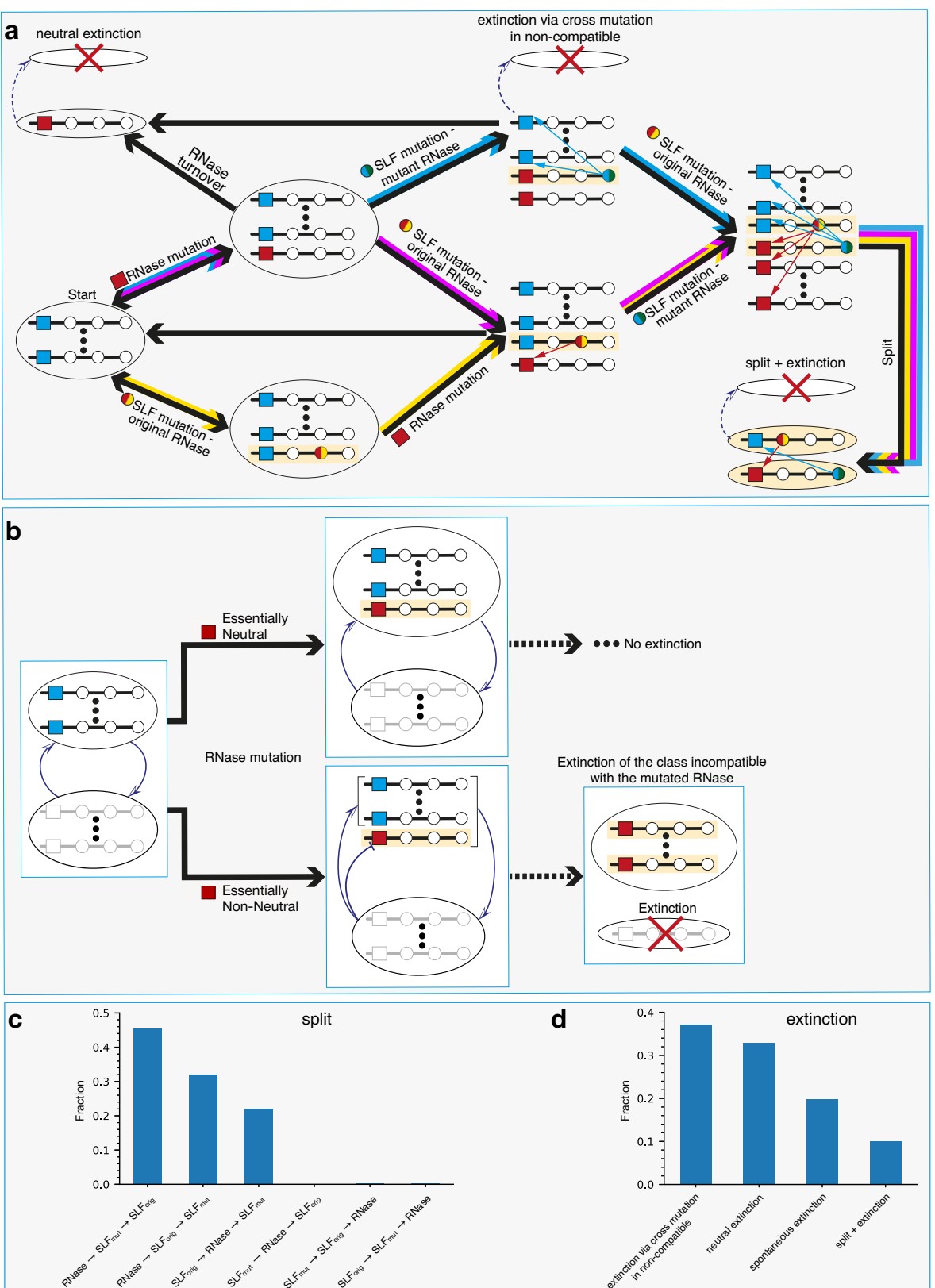

both male and female components[51], we find that unless pollen limitation is imposed, the haplotype dynamics is well-captured by the male-fitness alone. We illustrate the effect of pollen limitation in Fig. S23. The maximal male-fitness of a self-incompatible haplotype with $X_i$ copies is $1 - X_i/N$. If $X_i$ decreases, its fitness will increase, which would later lead to an increase in its copy number, and vice versa.

Hence, $\tilde{X}_i$ is a copy number predictor and averages out short-term fluctuations in $X_i$. Specifically, negative $\tilde{X}_i$ values predict the haplotype extinction. The maximal mean fitness of a $K$-class population is $(K-2)/K$, achieved if all classes have equal size (SI).

Following the first trajectory, we begin with the haplotype carrying the original RNase (blue horizontal bars). Then at the time marked

**Fig. 5 | The main class split and extinction trajectories are intertwined – a schematic diagram showing all the relevant transitions. a** All split trajectories require a sequence of three mutations: one RNase and two SLF mutations. The three split trajectories – here designated by the blue, magenta, and yellow paths – differ in the mutation order (see detailed description in the main text). A light yellow background highlights the surviving haplotypes that form the final classes after the split. If the RNase mutation is neutral, it could only lead to a class split. If, however, the newly emerging RNase mutation is incompatible as dam with one or more existing classes, it could also lead to the extinction of those classes in either of three ways, shown here as optional events (dashed arrows) accompanying the split pathways. Split is not obligatory even if the first and second mutations occurred. For example, following the RNase mutation, it is possible that the new RNase replaces the original one and the class returns to a single RNase state. Similarly,

after the second mutation, the advantageous haplotype carrying the SLF mutation could take over before the third mutation occurs. These transitions are shown by black arrows alone. **b** An "essentially neutral" RNase mutation (top) means that all foreign classes are either fully compatible as sires with the new RNase (red) similar to the original RNase (blue) or partially compatible but can adjust by changing their composition to regain full compatibility. An "essentially non-neutral" RNase mutation (bottom) means that at least one non-self class is (at least partially) sire-incompatible with the mutant RNase to the extent that it fails to adjust and restore full compatibility. For brevity, we omit below the word "essentially" and use the term neutral in this sense. **c, d** Occurrence histograms showing the fraction of each of the possible split (**c**) and extinction (**d**) trajectories, as observed in our simulations. In addition to the three extinction trajectories shown in (**a**), we also find "spontaneous" extinctions, occurring with no driving mutation, solely due to drift.

by $t_0$ a selectively neutral RNase mutation appears (red vertical bars). As a neutral mutation, the fitness of its carriers is equal to that of the original RNase haplotypes (note overlapping curves in the fitness plot). Its initial copy number is low, but it gradually grows neutrally. If it weren't for the SLF mutations, these two RNase variants could co-exist, until one of them would fix via drift. However, before such fixation happened, an SLF mutation, compatible as sire with the original RNase, occurred in one of the copies of the mutant RNase haplotype (red bar on a light blue circle). As the haplotype with this new SLF mutation has more siring opportunities than the other two haplotypes in its class (original RNase and mutant RNase with no SLF mutation), it immediately gains a fitness advantage over them, and its copy number sharply increases. The gray dashed line in the fitness plot (middle panel) shows the population mean fitness. Note that while the fitness values of original RNase and mutant RNase-only haplotypes fluctuate around this value, the new SLF-carrier fitness clearly supersedes the population mean. As all three haplotypes share a class (+appendix), an increase in one comes at the expense of the others. The initial surge of the new SLF mutant is due to the initially large copy number of the original RNase it is compatible with as sire. Later on, the decrease in the original RNase copy number reduces the SLF mutant's fitness advantage and moderates its expansion. Lastly, the third mutation occurs – an SLF mutation on the background of the original RNase – which grants compatibility with the new RNase. Similarly, the haplotype carrying this new mutation is advantageous compared to the original RNase-only carriers. Both the original RNase-only and mutant RNase-only carriers become extinct – note that $\bar{X}$ of these haplotypes go into the gray zone marking negative values. The vertical dashed line marks the time at which the daughter class completes its separation from the mother class. The new class then grows until it equilibrates with the existing ones.

The overall effect of all split trajectories is an increase in population mean fitness from $\frac{K-2}{K}$ to $\frac{K-1}{K+1}$ (observe the dashed line in the middle panel).

## Class extinctions are driven by incompatible RNase mutations

A newly emerging RNase mutation is not always essentially neutral. It might just as well be the case that it is incompatible as a dam with members of existing non-self classes (Fig. 5b), causing them a fitness deficit. If the new RNase expands, this could lead to the extinction of these class(es). Alternatively, since classes are genetically heterogeneous, it is also possible that if only a subset of a class is sire-incompatible with the new RNase, the class could change its composition accordingly, regain compatibility, and evade extinction. For extinction to occur, the new incompatible RNase does not necessarily need to fully replace the ancestral compatible RNase, but only to reach a sizable copy number (Fig. S14). The mutations mentioned earlier as part of class split trajectories, could just as well cause the extinction of another one, depending on the compatibility of the new RNase mutation. We identify three main extinction routes and illustrate them schematically in Fig. 5a. In Fig. 7 and Fig. S29, we show examples of the detailed dynamics and fitness of the

three main extinction trajectories, in a format similar to Fig. 6. Here however haplotypes from two different classes are shown. The colored symbols are associated with the class driving the extinction and the gray circles represent the class driven to extinction.

These extinction trajectories are the following. First, the incompatible RNase mutant could expand neutrally to a sufficient number, until it causes extinction ("neutral", Fig. S29). Second, the expansion of the incompatible RNase could be accelerated by an additional SLF mutation which confers compatibility with the original RNase ("cross mutation in non-compatible", Fig. 7). This is the most common path to extinction. Third, extinction could be concomitant to a split if the daughter class RNase is incompatible with another class ("split + extinction", Fig. S29). In addition, we also observe spontaneous class extinctions that are solely due to drift with no mutation involved and usually occur in very small classes (Fig. S18). In Fig. 5d, we show the frequencies of the different extinction trajectories, as observed in our simulations.

To assess the likelihood of either split or extinction following an RNase mutation, we quantified the number of non-self classes that newly emerging RNase mutations (surviving more than 100 generations) are incompatible with (Fig. S19). We found that >50% of these RNase mutations are compatible with all haplotypes of all non-self classes, hence splits are highly feasible.

While an RNase mutation could be incompatible with multiple classes, <13% of the RNase mutations are incompatible in the strict sense with more than one class (Fig. S19). Similarly, amongst the completed extinction events, we found that only 7% of the incompatible RNase mutations caused the extinction of two classes and only 0.6% caused the extinction of three classes, where the majority caused the extinction of only one class. Conversely, a given class could suffer from multiple incompatible RNase mutations occurring in parallel in different classes which collaboratively trigger its extinction. We find that 6% of the extinctions are caused by such parallel events. To conclude, the vast majority of extinction events is driven by a single RNase mutation and lead to the extinction of one class, which differs from the picture of mass class extinctions, previously proposed[29].

## Discussion

The SI locus is known to be highly polymorphic, but it remained obscure how this diversity evolved, since male and female-specificity determining genes encoded in this locus constrain each other's evolution, posing a "chicken-or-egg" problem. Here, we construct a model for the evolution of SI specificities ("S-alleles") in the CNSR SI system, which affords a more flexible interaction pattern between male and female specificities that alleviates some of the constraints imposed by previous models and uncovers new evolutionary routes.

Our evolutionary model employs a biophysical description of interactions between the RNase (female-determinant) and SLF (male-determinant) proteins. Such models afford several desirable properties. Most importantly, they let each protein have multiple distinct partners (Fig. S21), in agreement with empirical findings[11,33,52]. A biophysical description provides a natural notion of the distance

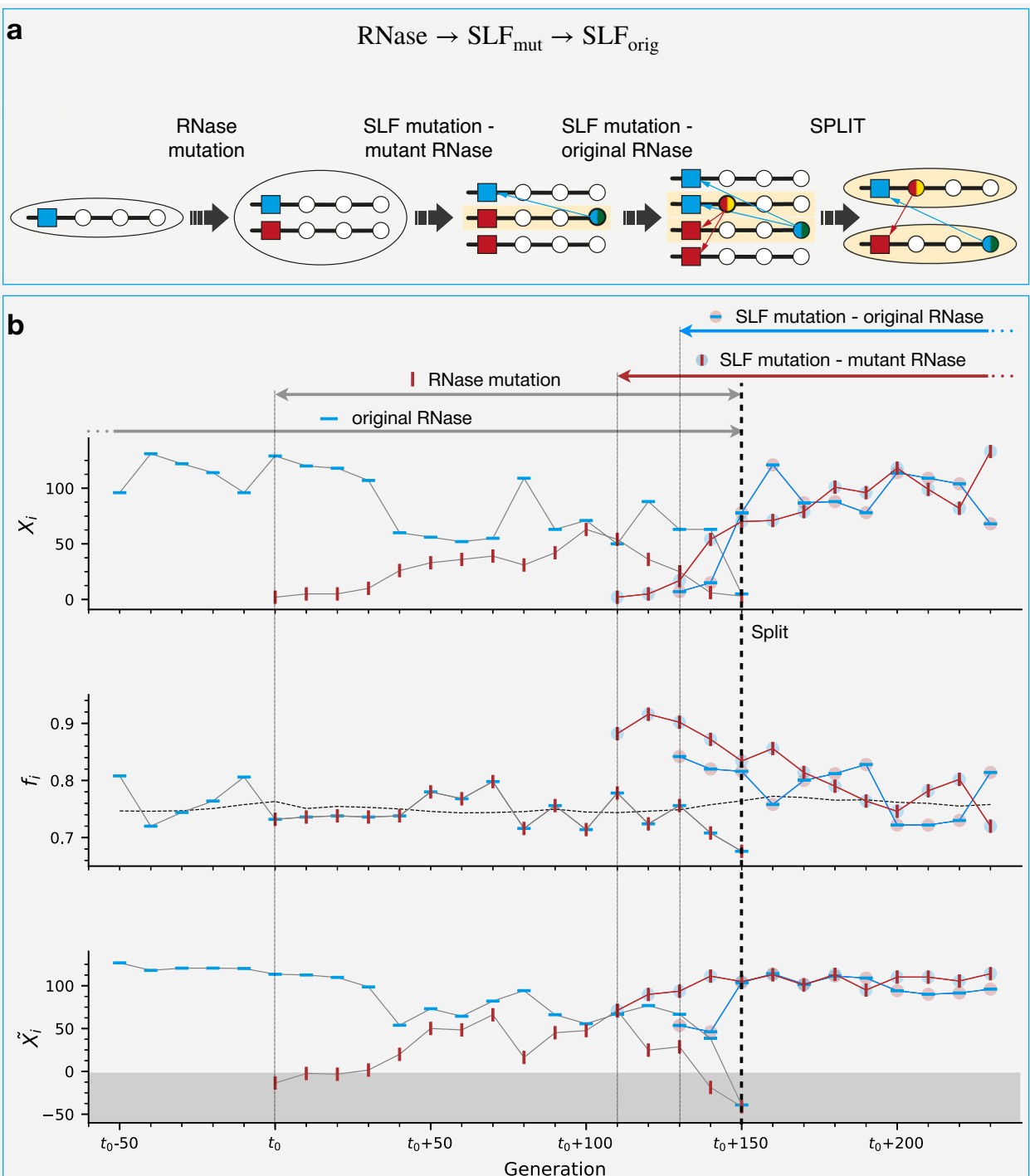

**Fig. 6 | Dynamics and fitness in the most common split trajectory. a** Schematic description of the sequence of mutations constituting the most common split trajectory (blue trajectory in Fig. 5). It starts with an RNase mutation followed by an SLF mutation on the background of the mutant RNase. Finally, a second SLF mutation compatible with the mutant RNase, which occurs on the background of the original RNase, gives rise to the daughter class separation from the mother class. Light yellow background designates the haplotypes that form the daughter classes. **b** An example of this trajectory from simulations. The three sub-figures show the copy numbers $X_i$ of the relevant haplotypes (top), their male-fitness $f_i$ (middle), and their fitness-adjusted copy number $\tilde{X}_i = X_i + (f_i - \bar{f}) \cdot N$, where $\bar{f}$ is the population mean male-fitness (bottom). The horizontal gray dashed line in the

middle sub-figure represents the population mean fitness. The gray-colored zone in the bottom sub-figure marks negative $\tilde{X}_i$ values. Haplotypes reaching negative values become extinct soon afterward. Different symbols represent different haplotypes with different combinations of mutations, as follows. The blue horizontal bar is the original RNase, red vertical bar is the mutant RNase. Colored circles surrounding the bars represent the SLF mutations, whereas the blue (red) circle represents an SLF mutation matching the original blue (mutant red) RNase. The arrows above mark the lifetime of each of these haplotypes. The vertical lines show the mutation occurrence times (solid lines) and the split time (thick dashed line). For simulation parameters see Table 1.

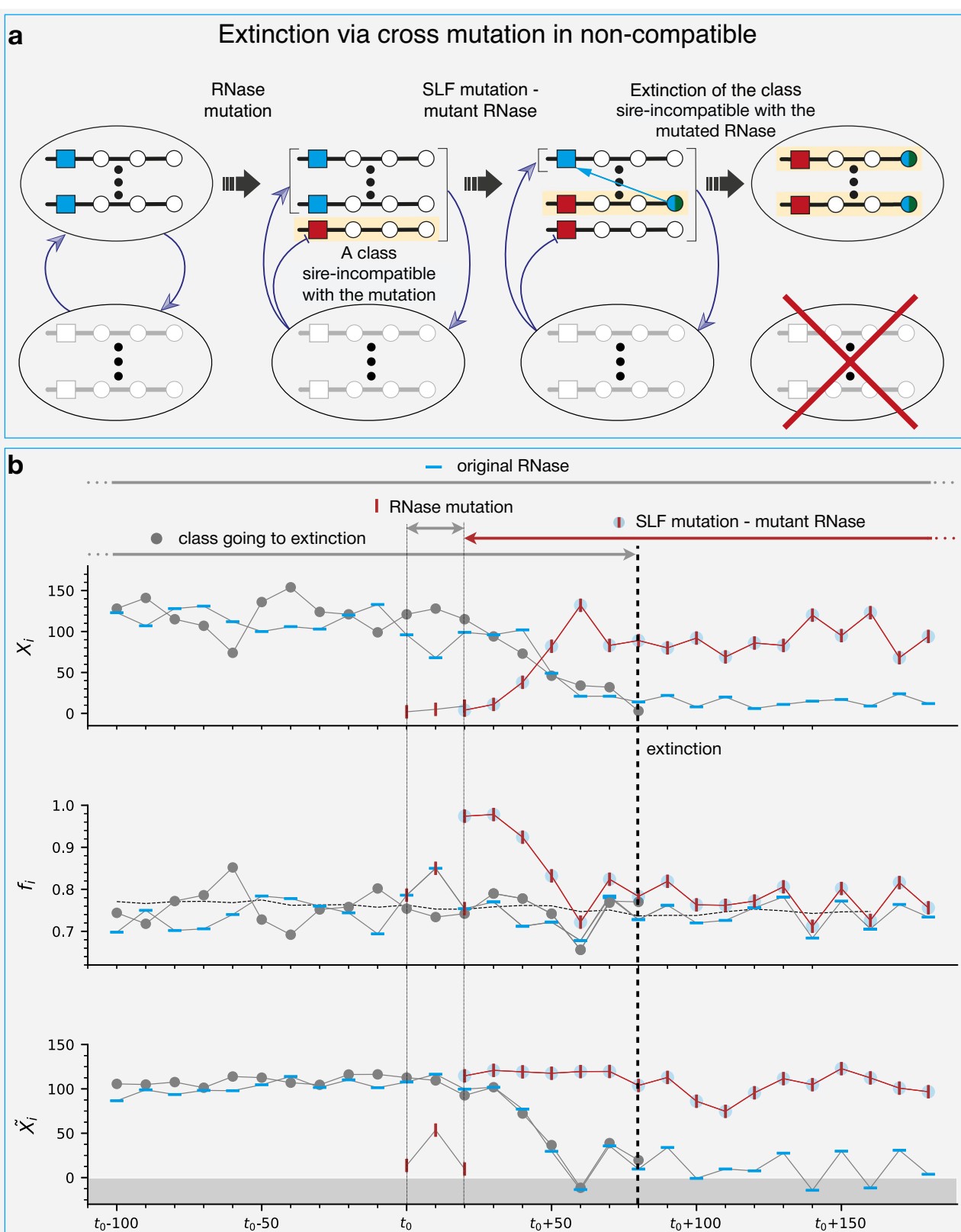

between distinct proteins, thus graduating the mutation space, allowing, for example, mutations that do not affect the compatibility phenotype[48,53]. It further allows for promiscuous recognition between arbitrary proteins, whose extent could be tuned by varying the interaction energy threshold $E_{th}$. Our definition of interaction between proteins induces correlation in compatibility phenotype between alleles similar in their sequences. Hence the compatibility phenotype of a mutant allele is highly correlated with that of its ancestor. This approach is in stark contrast to the more commonly used discrete-allelic models, in which only two values of distance between sequences were possible, and hence the description of mutation trajectories was over-simplified.

**Fig. 7 | Dynamics and fitness in the most common extinction trajectory.**
**a** Schematic description of the sequence of mutations constituting the most
common extinction trajectory (Extinction via cross-mutation in non-compatible in
Fig. 5). Similar to the split trajectory, it starts with an RNase mutation followed by an
SLF mutation on the background of the mutant RNase. Here, however, there is one
class (dimmed gray) whose members are incompatible as sires with the mutant
RNase. Following the expansion of this mutant RNase, that class becomes extinct
(red-crossed). **b** An example of this trajectory from our simulations. We use three
sub-figures, similar to Fig. 6, showing the copy numbers $X_i$ of the relevant
haplotypes (top), their male-fitness $f_i$ (middle) and their fitness-adjusted copy
number $\tilde{X}_i$ (bottom). Different symbols represent haplotypes with different
mutation combinations. The gray circles represent the doomed class. Other sym-
bols are as in Fig. 6. The gray dashed line marks the population mean fitness. The
arrows above mark the lifetime of each of these haplotypes. The gray-colored zone
in the bottom sub-figure marks negative $\tilde{X}_i$ values. Haplotypes reaching negative $\tilde{X}$
values become extinct soon afterward, as we see in this case for both the doomed
class and the original RNase of the driver class, which is replaced by the mutant
RNase. For simulation parameters see Table 1.

Our model's attractive properties facilitate several key dynamics
that were impossible in earlier models. Thanks to the inherent pro-
miscuity of interactions, it is likely that an arbitrary suite of SLF para-
logs is compatible with an unfamiliar RNase and even more likely if the
RNase is a mutant of an ancestral RNase the SLF-suite was compatible
with. Our model not only enables RNase mutations compatible with all
existing classes, except their own, but the vast majority of RNase
mutations are such (Fig. S19). Consequently, compatibility classes can
be genetically heterogeneous, not only in their SLF, but also in their
RNase content. Such neutral variation amongst RNases was indeed
detected in *Pyrus communis*[54] and a putative ancestral mutant pro-
posed to precede a split was demonstrated in *Solanum chacoense*[18].

A major outcome of our model is the spontaneous organization of
the vast majority of population haplotypes into compatibility classes,
such that there is incompatibility amongst class members, but full
compatibility across classes. We find three main paths to split and three
to extinction of classes, all of which require point mutations alone. None
of these paths pass through a self-compatible intermediate, nor rely on
gene conversion, as previously proposed[7,26,27,29–31]. Our two most pre-
valent split trajectories begin with a neutral RNase mutation, and hence
could not have been found in previous models, which disregarded such
mutations. Only our third split trajectory begins with a neutral male-
specificity mutation, similar to the fifth trajectory suggested by Bodova
et al.[31]. However, while in ref. 31 the neutral SLF mutation should have
occurred multiple times, one on each genetic background, in our model
it needs to appear just once, thanks to promiscuity. Trajectories tra-
versing through a self-compatible intermediate are also possible in our
model, but are much less common (Fig. 5c and Fig. S17).

All split trajectories require a sequence of three mutations, where
the first (RNase or SLF) is essentially neutral. The second mutation
confers a fitness advantage to a sub-class ("driver") allowing it to cross
another sub-class ("endangered"). The endangered sub-class could
vanish unless it gains a rescue mutation allowing it to cross back the
driver sub-class and restore its fitness. We find that the time window
for the rescue mutation allows for sufficient opportunities for split
completion in a wide parameter range. This is partly because the fit-
ness advantage of the driver sub-class is not constant but rather
depends on the size of the endangered sub-class. Hence, this fitness
difference is maximal upon mutation occurrence but later vanishes as
the endangered population declines and the driver expands. In the
absence of the third ("rescue") mutation we distinguish two phases in
the sub-class dynamics following the second mutation. In the first
phase, the driver is rare, hence its fitness advantage is significant and
its numbers are rising while the endangered sub-class is declining. In
the second phase, the driver is already prevalent and the endangered is
on the verge of extinction, hence the fitness difference between them
is negligible. Though unstable, the second phase allows additional time
to gain the rescue mutation. For more details see mathematical ana-
lysis in Supplementary Information.

We find that class extinction is driven by an RNase mutation
incompatible with another existing class, in agreement with previous
models[29,31]. Yet, due to the promiscuity of interactions, most RNase
mutations maintain compatibility with existing non-self classes. RNase
mutations that do not are mostly incompatible with only a subset of
the class haplotypes (Fig. S19), allowing for class recovery. Thus class

extinctions typically occur one at a time, in contrast to the expecta-
tions of previous models[29,30].

Classes continuously emerge and decay, but the ratio between
extinctions and splits varies with the number of existing classes, such
that for low class number splits predominate, but for high number
extinctions do. Thus, the class count fluctuates around a stable equi-
librium, whose value positively correlates with the population size,
mutation rate, and $E_{th}$. We conjecture that extinction probability
increases with the number of classes because it is affected by within-
class SLF diversity (smaller classes are less diverse and hence less
resilient to new RNases) and the higher number of classes that should
match each new RNase. Hence, the higher the number of classes, the
lower the probability that a new RNase mutation is neutral. In contrast,
if classes are fewer and hence larger, they offer both more targets for a
rescue mutation and a longer time for its occurrence, because the
driver sub-class takes longer to spread. Both factors increase the
chances for split completion before the endangered sub-class vanishes
(1st and 2nd trajectories). Larger classes also exhibit higher SLF
diversity, hence a new RNase mutation is more likely to already have a
compatible SLF in its own class (3rd trajectory) (Figs. S15, S16). This
explanation deviates from the selection-drift balance which was pro-
posed to determine the number of specificities in SR systems[3,55–57]
implying that a class needs to reach a minimal size to survive. In con-
trast, we argue that the primary barrier to class number increase is not
class size, but the requirement that each class should be compatible
with all others, hence a population can only tolerate a limited number
of classes. Extinctions due to drift, mostly of very small classes
(Fig. S18), do occur in our model but are only secondary (Fig. 5d).

Our model is robust and our main results hold under a broad
range of parameter values, including varying $E_{th}$, number of SLF
paralogs per haplotype, population size, mutation rate, and various
levels of inbreeding depression. Specifically, we have verified that
equilibrium in the number of specificities is obtained under various
parameter values (Figs. S4–S13), as long as interaction between pro-
teins is rather likely. Similar to previous models[29,31], maintenance of SI
requires high inbreeding depression $\delta$ (Figs. S24–S26) and high self-
pollen supply $\alpha$[58]. These parameters affect the relative proportions of
the different diversification trajectories, and specifically under lower
inbreeding depression, trajectories passing through a self-compatible
intermediate become more common, yet remain second to the three
trajectories maintaining SI throughout (Fig. S17).

Further investigation of the model properties should address the
quantitative relation between the population size and mutation rate to
the number of classes, their typical lifetime, and within-class diversity.
It is also interesting to determine whether the current model could be
further simplified while still maintaining its current behavior.

Linking our model to population genetic and genomic data should
be very insightful and allow testing some of the model's qualitative
predictions, as well as refining the model and calibrating its para-
meters. These include estimation of the number of residues distin-
guishing between different RNase alleles[17] and functionally different
SLFs[19], and assessment of between- and within-haplotype SLF
diversity[21,59]. Our model offers predictions regarding the within-class
allelic diversity and its dependence on the class size, which could then
be contrasted with genomic data. We find that the number of classes

should increase with population size (Figs. S6–S8), in line with previous data for SR[60]. Yet, as the quantitative relation between the population size and number of specificities could vary between SR and CNSR, it would be insightful to examine this for CNSR population data as well.

The number of SLF paralogs per haplotype determined empirically for species with CNSR SI were 16–20 for *Pyrus*[61], *Malus*[21], and *Petunia*[62,63], but 32 for *Antirrhinum*[59]. The number of distinct RNases found in natural population surveys was 20 for *Sorbus aucuparia*[64] and 25 for *Pyrus pyraster*[65], but these may be under-estimates due to limited sample sizes. The number of SI specificities in a species is thought to exceed the number of per-haplotype SLF genes: for example, in *Malus* 34 specificities were identified, but only 17–19 SLFs per haplotype[21]. The number of compatibility classes in our model (8–9 under baseline parameters) is slightly smaller than the number of SLF sequences per haplotype (on average 10 – Fig. S20) though. A possible reason for this discrepancy could be spatial heterogeneity in natural populations, such that SLFs evolve to only recognize RNase specificities in their vicinity, but not necessarily more distant ones, whereas our model assumes a panmictic population.

The number of compatibility classes in our model positively correlates with the population size and $E_{th}$ (determining the probability of proteins to interact at random) and also increases under extreme pollen limitation (Fig. S23) as previously found[51]. With $250 \leq N \leq 1000$ and $-8 \leq E_{th} \leq -4$ the equilibrium number of classes we obtained ranged between 6 and 13 (Figs. S4–S8). The larger number of classes found in natural populations could be explained by their larger sizes. Additionally, we hypothesize that population spatial structure, such that each individual has access only to a limited number of mating partners in its neighborhood, could support an even greater allelic heterogeneity beyond our current findings.

Previous models for the number of SI specificities in finite populations assumed SR[3,57,66]. These models provided higher estimates of the number of specificities compared to the number of classes obtained in ours (~12 specificities for $N = 500$, $\mu = 10^{-5}$ compared to 8–9 classes in our case). We hypothesize that this discrepancy is rooted in the different implications on SR vs. CNSR of the requirement that every specificity is compatible with all others. A CNSR specificity must maintain multiple compatibilities with all other specificities, a burden that becomes heavier the more specificities exist already. In contrast, an SR specificity only needs to be distinguishable from existing ones, but not to be able to recognize each and every one of them, hence the emergence of novel specificities is much less decelerated with the growing number of existing ones.

Recent experiments demonstrated that an RNase from *Petunia* could be detoxified by various SLFs from different species or even different families[35]. This surprising finding is congruous with the promiscuity of interactions in our model (Fig. S21). We also predict that RNase and SLF proteins mutated at their interaction interfaces are even more likely to interact with their pre-mutation partners than with random ones. It would be very interesting to further test this experimentally.

How do proteins maintain interactions with multiple distinct partners? Do they interact with all partners via a shared interface or do they have separate or partially overlapping interfaces for different partners[40]? Here we assumed a single interaction interface per protein and did not consider shifted binding, yet it is possible that different positions in the binding interface specialize in different partners. We hope that future structural studies of these proteins will shed light on this question, and future refinement of our model could include this aspect, too. Other simplifications in our model include the lack of gene conversion and the assumption of a well-mixed population with no spatial structure. Future extensions of our model should include these as well.

Which factors determine the size of biological networks is a fundamental question applicable to different biological networks. Biochemical interactions between molecules often have relatively low energies and hence are inherently promiscuous. This promiscuity is a double-edged sword, where functional fidelity preservation could limit the network size. Indeed, avoidance of non-specific protein–protein interactions was previously proposed to limit the total number of co-existing proteins in a cell[39,50] and regulatory crosstalk was hypothesized to limit the size of gene regulatory networks[67,68]. Yet, promiscuity also grants the network additional degrees of freedom rendering it more evolvable[48,69,70].

In summary, we present an evolutionary-biophysical model for the evolution of mating specificities in the CNSR SI system. Our model is the first to allow for multiple interactions per molecule. It deciphers the evolutionary trajectories of SI specificity birth and death and being more biologically realistic offers connections to genomic data. It highlights the role promiscuous molecular recognition plays in determining the network size. Similar promiscuity may be shared by additional biological networks and the framework proposed here can be used to address further questions regarding network growth and complexity.

## Methods

### Detailed description of the simulation flow

We constructed a population-level stochastic simulation, to study the evolution of SI specificities in the CNSR SI system. The simulation data analysis scripts were written in Python and run on a local server. We summarize the notation and default parameter values used in Table 1.

**Data structure.** We consider a population of $N$ diploid individuals, composed of two haplotypes each. Every haplotype contains a series of genes: a single RNase encoding the female-specificity, and a variable number of SLF paralogs, encoding the male-specificity.

Each gene (RNase or SLF) is represented by a sequence of $L$ amino acids. Each amino acid belongs to one of four biochemical categories (hydrophobic, neutral polar, positively charged, negatively charged). We draw the sequences from the prior frequencies $p_{prior} = [0.5, 0.265, 0.113, 0.122]$, as obtained from the UNIPROT database (https://www.uniprot.org/) – see Table S1.

Our stochastic simulation was initiated with a population of $N$ diploid individuals, each comprised of two haplotypes (described below).

**Interaction energy between pairs of amino acids.** Interaction energies between all pairs of amino acids are summarized in Table 2.

**The Population Initiation module.** To initialize the population we first constructed a complete set of $n_h$ self-incompatible haplotypes, such that each haplotype is bidirectionally compatible (both as male and as female) with all others. We constructed this initial set of $n_h$ haplotypes by the following two steps:

1. *Draw RNase sequences*. Draw an entire haplotype (one RNase and $n_h - 1$ SLFs). If this haplotype is not self-compatible, namely, if none of the SLFs is compatible with the RNase – keep it. Repeat this procedure until $n_h$ self-incompatible haplotypes are drawn. Then, remove the SLFs and keep only the $n_h$ RNase sequences of these haplotypes. SLFs will be drawn again separately. This procedure ensures symmetry between the various RNases, such that neither is more or less constrained than the others.

### Table 2 | Interaction energies between all pairs of amino acids

|   | H | P | + | − |
|---|---|---|---|---|
| H | −1 | 0 | 0 | 0 |
| P | 0 | −0.25 | −0.25 | −0.25 |
| + | 0 | −0.25 | 1 | −1.25 |
| − | 0 | −0.25 | −1.25 | 1 |

2. *Draw SLF sequences with one-to-one RNase–SLF match*. Draw a single SLF sequence. If this SLF sequence is compatible with exactly one RNase from the previously drawn RNase set, place this SLF on all haplotypes except for the one whose RNase it matches. Continue this procedure, until $n_h$ SLF sequences were found, such that each is compatible with exactly one unique RNase.

The outcome is a set of $n_h$ complete SI haplotypes, such that each haplotype contains a single unique RNase and a set of $n_h - 1$ SLFs that are compatible with the remaining $n_h - 1$ RNases. $n_d = n_h \cdot (n_h - 1)/2$ distinct diploid heterozygous genotypes can be constructed from these $n_h$ haplotypes (homozygous genotypes are impossible, because all haplotypes are self-incompatible). We then draw $N$ diploid heterozygous genotypes with equal probabilities.

**The simulation life cycle.** The next four steps repeat every generation: gene duplication, gene deletion, mutation, and construction of the next generation, as detailed below.

1. *Gene duplication*. Each SLF (but not RNase) sequence can be duplicated with probability $p_{dup}$ per generation. The duplication was simulated by adding an identical copy of the gene to the haplotype sequence.
2. *Gene deletion*. Each SLF (but not RNase) sequence can be deleted with probability $p_{del}$ per generation.
3. *Mutation*. Each amino acid in any of the genes (RNase or SLF) can be mutated with probability $p_{mut}$ per generation. The values of amino acids chosen to be mutated are re-drawn from the a priori amino acid bio-class frequencies $p_{prior}$ = [0.5, 0.265, 0.113, 0.122].
4. *Construction of the next generation*. Compute $P_{\vec{r}}$, the probabilities to obtain all possible diploid genotypes $\vec{r}$ in the next generation, given their frequencies in the current generation $x_{\vec{r}}$, the compatibility relations between all pairs of maternal diploid genotype $\vec{g}$ and haploid $h$ and Mendelian segregation rules, namely which offspring genotype could be produced by each parental combination (using similar notation to ref. 31):

$$P'_{\vec{r}} = \frac{1}{\bar{W}} \left[ (1 - \delta) \sum_{\vec{g}} x_{\vec{g}} \pi^{self}_{\vec{g} \leftarrow \vec{g}} R_{\vec{g}, \vec{g} \rightarrow \vec{r}} + \sum_{\vec{g}} x_{\vec{g}} W_{\vec{g}} \sum_h \pi^{out}_{\vec{g} \leftarrow h} R_{\vec{g}, h \rightarrow \vec{r}} \right] \quad (2a)$$

$$\bar{W} = (1 - \delta) \sum_{\vec{g}} x_{\vec{g}} \pi^{self}_{\vec{g} \leftarrow \vec{g}} + \sum_{\vec{g}} x_{\vec{g}} W_{\vec{g}} \sum_h \pi^{out}_{\vec{g} \leftarrow h} \quad (2b)$$

where the first term represents offspring due to self-fertilization and is non-zero only if self-compatible genotypes exist. Even if possible, these offspring survive with a lesser probability of $1 - \delta$ relative to offspring resulting from outcrossing. The second term represents offspring due to outcrossing.

$R$ represents the probability that diploid offspring genotype $\vec{r}$ is produced by a certain parental combination $\vec{g} \leftarrow \vec{g}$ or $\vec{g} \leftarrow h$ following the Mendelian segregation rules, where $R_{[\vec{g}, \vec{g} \rightarrow \vec{r}]}$ refers to self-fertilization with identical pollen and pistil genotype $\vec{g}$, and $R_{[\vec{g}, h \rightarrow \vec{r}]}$ refers to outcrossing, with arbitrary diploid pistil genotype $\vec{g}$ and haploid pollen genotype $h$, respectively:

$$R_{[\vec{g} \leftarrow \vec{g}] \rightarrow \vec{r}} = \begin{cases} 1 & \vec{g} = [S_i, S_i] \text{ is full SC and } \vec{r} = [S_i, S_i] \\ 0.5 & \vec{g} = [S_i, S_j] \text{ is full SC and } \vec{r} = [S_i, S_j] \\ 0.25 & \vec{g} = [S_i, S_j] \text{ is full SC and } \vec{r} = [S_i, S_i] \\ 0.5 & \vec{g} = [S_i, S_j] \text{ is half SC (only } S_i) \text{ and } \vec{r} = [S_i, S_j] \\ 0.5 & \vec{g} = [S_i, S_j] \text{ is half SC (only } S_i) \text{ and } \vec{r} = [S_i, S_i] \\ 0 & \text{otherwise.} \end{cases} \quad (3a)$$

$$R_{[\vec{g} \leftarrow h] \rightarrow \vec{r}} = \begin{cases} 1, & \vec{g} = [S_i, S_i], h = S_i \text{ and } \vec{r} = [S_i, S_i] \\ 1, & \vec{g} = [S_i, S_i], h = S_j \text{ and } \vec{r} = [S_i, S_j] \\ 0.5, & \vec{g} = [S_i, S_j], h = S_j \text{ and } \vec{r} = [S_i, S_j] \\ 0.5, & \vec{g} = [S_i, S_j], h = S_j \text{ and } \vec{r} = [S_j, S_j] \\ 0.5, & \vec{g} = [S_i, S_j], h = S_k \text{ and } \vec{r} = [S_i, S_k] \\ 0.5, & \vec{g} = [S_i, S_j], h = S_k \text{ and } \vec{r} = [S_j, S_k] \\ 0, & \text{otherwise.} \end{cases} \quad (3b)$$

Note that outcrossing could potentially occur between genetically identical pollen and pistil, but could give rise to offspring only if they are self-compatible. $\pi^{self}_{\vec{g} \leftarrow \vec{g}}$ and $\pi^{out}_{\vec{g} \leftarrow h}$ represent the probability that a maternal plant with genotype $\vec{g}$ is either self-fertilized or fertilized by pollen genotype $h$, respectively,

$$\pi^{self}_{\vec{g} \leftarrow \vec{g}} = \frac{\alpha \Delta^D_{\vec{g} \leftarrow \vec{g}}}{\alpha \Delta^D_{\vec{g} \leftarrow \vec{g}} + \sum_{h'} (1 - \alpha) x_{h'} \Delta^H_{\vec{g} \leftarrow h'}} \quad (4a)$$

$$\pi^{out}_{\vec{g} \leftarrow h} = \frac{(1 - \alpha) x_h \Delta^H_{\vec{g} \leftarrow h}}{\alpha \Delta^D_{\vec{g} \leftarrow \vec{g}} + \sum_{h'} (1 - \alpha) x_{h'} \Delta^H_{\vec{g} \leftarrow h'}} \quad (4b)$$

accounting for the current population frequencies of the various pollen haploid genotypes $x_h$, the proportion $\alpha$ of self-pollen received, and the pollen–pistil compatibilities $\Delta$:

$$\Delta^D_{\vec{g} \leftarrow \vec{g}} = \begin{cases} 1, & \text{if } \vec{g} \text{ is fully self-compatible} \\ 0.5, & \text{if } \vec{g} \text{ is half self-compatible} \\ 0, & \text{if } \vec{g} \text{ is self-incompatible} \end{cases} \quad (5a)$$

$$\Delta^H_{\vec{g} \leftarrow h} = \begin{cases} 1, & \text{haploid pollen } h \text{ is compatible with diploid maternal genotype } \vec{g} \\ 0, & \text{haploid pollen } h \text{ is incompatible with diploid maternal genotype } \vec{g} \end{cases} \quad (5b)$$

A diploid maternal plant is considered half self-compatible if exactly one of its two haplotypes is self-compatible, but the other is not. It is fully self-compatible if both haplotypes are self-compatible. Otherwise, it is self-incompatible. Once we calculated all the probabilities $P_{\vec{r}}$, we randomly draw the genotypes of $N$ diploid offspring using these probabilities. This offspring population then replaces the parental population, which completes as full generation. We then return to step (1).

To easily keep track of the possible crosses in the population we form a table of all sire-dam pairings (including self-fertilizations) and determine their compatibilities, based on their allelic contents. Every generation we update this table as necessary to account for the effect of newly emerging mutations on fertilization capabilities.

**Number of fertilization attempts.** While pollen grains are typically more abundant than ovules, it is still possible that certain maternal plants suffer from pollen limitation, for example, if they possess a female-specificity with too few compatible sires. To account for that we modified the fitness function to reflect a limited number $k$ of fertilization attempts per maternal plant:

$$W_{\vec{g}} = 1 - (1 - y_{\vec{g}})^k \quad (6a)$$

$$y_{\vec{g}} = \frac{\sum_h x_h \Delta^H_{\vec{g} \leftarrow h}}{\sum_h x_h} \quad (6b)$$

where $y_{\vec{g}}$ is the proportion of pollen that is compatible with diploid maternal genotype $\vec{g}$ out of all foreign pollen the maternal plant receives. In practice, as our model works in a regime, where the vast

majority of novel RNases are compatible with all existing pollen genotypes, this parameter had very little effect, unless extreme pollen limitation was assumed ($k = 1$ to $2$).

### The simulation data structure and output

We store the population state in the HAPLOTYPE data structure, which includes a list of haplotype objects. Each object in the list contains the following information: the haplotype's current copy number, the identity of the haplotype ancestor at the initial condition, the generation in which the haplotype first emerged, the identity of its direct parent haplotype, and its allelic content, namely the list of SLFs and RNase indices it contains. This data structure only keeps the information of the current population and is updated every generation. Gene sequence indices are set uniquely, such that every new variant receives a unique index, not given previously to any other one, including sequences that no longer exist in the population. If the same sequence emerged independently more than once, it received the same index upon each emergence. Furthermore, the simulation keeps a list associating the gene sequence indices with the relevant amino acid sequences for all SLFs and RNases that ever existed in the population. In addition to the current population state stored in the HAPLOTYPE structure, we also keep lists of the parental indices for all haplotypes that ever existed in the population starting from the time that the entire population descends from only one of the initial haplotypes. This information is essential when tracking the class evolution (see below) and especially detecting the split/extinction moment.

During each simulation run, we save the HAPLOTYPE data structure documenting the population state, every 10 generations for further offline analyses. We analyze the simulation data only from the time point that all the population haplotypes descend from only one of the $n_h$ ancestral initial haplotypes. By that, we minimize the dependence of the results on the specific initial condition of each run. The typical run-time needed until only a single ancestor remained was ≈50,000 generations. From that time point the simulations were run for additional 100,000 generations.

### The haplotype classification algorithm

Every 10 generations, we classified the set of haplotypes in the population according to their compatibility phenotype. The classification should obey the following two requirements: a haplotype affiliated with a class should not be compatible neither as sire nor as dam with any of the other members of the same class, but should be compatible both as sire and as dam with all members of all other classes, it is not affiliated with. Following this definition, it is not guaranteed that all haplotypes are classified into any of the classes. Specifically, self-compatible haplotypes cannot be associated with any class, because they can fertilize their own class members. The division of haplotypes into classes is not unique. To ensure that the most frequent haplotypes are classified, we first ordered all the self-incompatible haplotypes in the population in descending order of their copy number. We then defined the most frequent haplotype to be the first class. Then, for every unclassified haplotype in its turn (following the descending copy number order), we tested its compatibility with the already existing classes. If it was bidirectionally incompatible with all members of exactly one class and bidirectionally compatible with all members of all the remaining classes, we associated it with the class it was incompatible with. If it was bidirectionally compatible with all members of all existing classes, we defined a new class and associated this haplotype with that class. Otherwise, this haplotype remained unclassified and we moved on to the next one. Importantly, our class definition refers only to the compatibility phenotype of haplotypes, rather than to their genotype. As our model allows for multiple partners per RNase/SLF, classes could be genetically heterogeneous.

This classification procedure was repeated every 10 generations independently of previous classifications. As haplotypes mutated during the simulation, this could influence their class association. For example, if a haplotype gained an SLF mutation that rendered it compatible as a sire with (some of) its class members, it no longer followed the requirement of being incompatible with all its class members. Similarly, if an RNase mutation occurred, that was incompatible as a dam with members of foreign classes, it no longer followed the requirement for bidirectional compatibility with all non-self classes. Many of these mutations have a short lifetime, resulting in haplotypes that exit and enter the classes frequently. Such mutations are also part of the trajectory to class split and extinction. Hence, to keep the information regarding the class from which the (temporarily) unclassified haplotype originated, and the one to which it could soon return, we additionally defined class appendices.

These class appendices include former class members that following a mutation, either lost compatibility with members of foreign classes or gained compatibility with their class members or both. Hence, each haplotype is uniquely associated with either one of the classes or with one of the class appendices, but never with two of them. This definition of class appendices allows us to conveniently track class evolution.

### Identification of split and extinction events

Other possible changes in the population class structure are the split of an existing class into two separate ones and the extinction of an entire class. To identify such events we need to associate between the class structures obtained at different time points. To accomplish that, we tracked the parents in the previous classification time point of haplotypes affiliated with a certain class. If all the parents of haplotypes in that class are also affiliated with one class in the previous classification, we identify the current class with the parental class. We identify a split if the offspring of parents associated with one class in the previous classification are associated with two separate classes in the current classification. Such a split is the end of the process during which, mutated haplotypes, containing mutated RNase/SLFs, were associated with the class appendix.

Class extinction was defined as the event that all haplotypes of a class and its appendix leave no descendants in the next classification, including no mutants of the former class haplotypes. We specifically require that the appendix members also vanish since a haplotype can alternate between being associated with a class and being associated with its appendix, depending on the state of the remaining classes.

### Haplotype grouping

In Figs. 6, 7, we illustrate copy numbers and fitness of haplotypes involved in split and extinction events. There are five categories of haplotypes shown there: "original RNase", "mutant RNase", "original RNase + SLF", "mutant RNase + SLF", and "extinct RNase" (see the main text for full details). Yet, each category could potentially contain multiple different haplotypes. For example, the "original RNase" category refers to all haplotypes sharing that RNase, but they could vary in their SLFs content, as long as this has no implication on haplotype compatibility. Similarly, some "mutant RNase" copies could later acquire SLF mutations that do not affect their functionality. For simplicity of presentation, we group together all haplotypes sharing a common functionality and show the sum of their copy numbers.

**Calculation of fitness-adjusted copy number for grouped haplotypes.** For ease of presentation, we lump together distinct haplotypes with shared functionality (e.g., in Figs. 6 and 7). The fitness-adjusted copy number is defined as $\tilde{X}_i = X_i + (f_i - \bar{f}) \cdot N$, where $X_i$ is the number of copies, $f_i$ is the proportion of diploid maternal plants it is compatible with as sire and $\bar{f}$ is the population mean fitness. If these different haplotypes happen to have the same fitness value, we simply add their copy numbers. For example, for two haplotypes $X_1$, $X_2$ with equal

fitness values $f_{1+2} = f_1 = f_2$:

$$\tilde{X}_{1+2} = X_1 + X_2 + (f_{1+2} - \bar{f}) \cdot N. \qquad (7)$$

If however, they differ in their fitness, we use in the formula the weighted average of their fitness values

$$f_{1+2} = \frac{f_1 \cdot X_1 + f_2 \cdot X_2}{X_1 + X_2} \qquad (8)$$

## Reporting summary

Further information on research design is available in the Nature Portfolio Reporting Summary linked to this article.

## Data availability

The authors declare that all data supporting the findings of this study are available within the article and its Supplementary Information file.

## Code availability

The code is available on gitHub repository https://github.com/Tamar-Friedlander-Lab/evol-RNase-based-SI_stochast_simul.

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

## Acknowledgements

We thank Idan Efroni, Yonatan Friedman, and Avi Mayo for their comments on the manuscript, and Amnon Horovitz for useful discussions. This research was supported by the Israel Science Foundation, Grant Number 1889/19 (T.F.).

## Author contributions

T.F. conceptualized the project, acquired funding and supervised. K.E and A.J. wrote and ran the simulation, collected and analyzed results, and prepared figures. O.F. contributed the mathematical analysis. T.F and O.F developed the methodology and wrote the paper.

## Competing interests

The authors declare no competing interests.
