## [Peer Review File · Nature Communications]

The role of promiscuous molecular recognition in the evolution of RNase-based self-incompatibility in plantsREVIEWER COMMENTS

Reviewer #1 (Remarks to the Author):

The manuscript by Erez et al. tackles the question of the evolutionary enigma of evolution of new specificities in self-incompatible plants with a collaborative non-self recognition (CNSR) system, such as the S-RNase system of Solanaceae. This theoretical study introduces a very original model that allows for the first time to implement promiscuous molecular recognition between male and female genetic components, a well-known property of CNSR systems which had not been included in theoretical models up to now. It is shown that allowing for promiscuous interactions changes several fundamental outcomes of the models, including evolutionary stability of the number of allelic specificities (which was unclear in previous models), and the introduction of new evolutionary paths that were previously excluded. The implementation of the model and its application are well described overall (see below for some problems of clarity), although it requires very careful attention of the reader to grasp all details. Maybe the authors could find ways to present the model with two levels of description, one simplified presentation highlighting the main results, and one deeper presentation showing all the details. Overall, this study is providing a very important contribution to the understanding of the evolution of CNSR systems. However, beside the question of clarity raised above, the manuscript, in its current version, also does not adequately tackle the following issues, which should be addressed in a revision:

1. Comparison of the observed number of allelic classes as a function of population size with previous theoretical models: here the authors find that the average expected number of allelic classes is about 8-9 for a population size $N=500$. This could have been compared to the classical expectations for Self-Recognition (SR) systems under gametophytic SI produced by Wright and followers. For instance, using Yokoyama and Nei (1978, Genetics) equations, the number of common allelic lines expected with $N=500$ is ranging from $n_a=12,2$ to $n_a=10,3$ for allelic mutation rates varying from 10^{-6} to 10^{-8} . Hence, allele numbers expected under CNSR seem to be lower than under SR. This could be discussed, and could allow some relevant comparisons with empirical results because gametophytic systems with either CNSR (Solanaceae, Plantaginaceae, Maloideae within Rosaceae) or SR (Papaveraceae, Prunus within Rosaceae) have been studied.

2. Negative frequency-dependent selection acting on the female fitness (Fecundity selection): without explicit recognition, the authors completely obliterate selection acting on female fitness because their implementation, in case of incompatibility between randomly chosen parents "grant multiple opportunities to mate with uniformly chosen non-self pollen grains". Hence, they neglect situations with pollen limitations, although these are often found in nature (Knight et al 2005, Ann Rev Ecol Evol Syst). It has been shown that including pollen limitation in theoretical models of SI evolution can greatly impact the evolutionary outcome as it adds an additional component of negative frequency-dependent selection, coined "fecundity selection" (Vekemans et al. 1998 Evolution). With respect to the model presented here, fecundity selection would potentially greatly impact the evolution of S-RNase mutants (and thus the most frequent scenario in this study where a S-RNase mutation occurs first) because they will suffer from lower (female) fitness if they become incompatible with one or more of the previous allelic classes (in this study, such mutations are assumed to be almost neutral, but this would not be true under pollen limitation models). They recognize that the S-RNase mutant could be non-neutral but in this case they predict a reverse outcome (lower fitness and putative extinction of the other allelic class and not of the S-RNase mutant). It would be important to test the effect of pollen limitation, or at least to discuss this issue.

3. Expected level of molecular promiscuous interactions: the originality of the model is that it allows promiscuous interactions, i.e. a single SLF molecule can recognize (and detoxify) several S-RNase molecules, including some that are not (yet) present in the population. This property is the basis of the original results found in the outcome of the model, as compared to previous models. However, it is frustrating that the authors do not present results on the distribution of this overlap of functional recognition. This information seems to me to be crucial to understand properties of the model, although obviously it will depend on model parameters. It is also very interesting because it could be compared with empirical estimates. Such estimates are still rare in the literature but many groups are currently investigating molecular approaches to tackle this issue empirically (with functional tests). Hence, producing expectations for the average number S-RNase alleles recognized by a single SLF molecule, and for the variance in overlap between different SLF molecules, would be very useful. I don't understand why this has not been addressed by the

authors.

4: Variation in the number of SLF sequences per S-haplotype: in the model, diversity in the interaction properties of SLF sequences can arise because of duplication and mutation or because of mutation only. The variability in the number of SLF sequences per S-haplotype is of interest, and again especially because it could be compared with empirical observations. It would be interesting to have expected results about this variability, and to discuss those.

5: Also of interest, would be to compute the level of "completeness" (the proportion of S-haplotypes that can detoxify, based on their set of SLF sequences, all members of all other classes) of S-haplotypes in the population, as this has been shown to be an important variable in previous models.

6. Full inbreeding depression: in most simulations, the inbreeding coefficient is set to 100%. Hence, under these circumstances all scenarios with a self-compatible intermediate are completely disfavored (see figure 6C). I think this should be more explicitly discussed, as this artificially increases the contrast in outcome of this study as compared to previous studies (using full inbreeding depression will favor scenarios that do not lead to self-compatible intermediates).

7. Reference to the literature on dual-specificity S-RNase mutants: Matton et al 1999 (Plant Cell) found in *Solanum chacoense* that by experimentally altering amino acids of an S-RNase sequence, they generated a so-called "dual-specificity" mutant that behaves initially similarly to its parental allele (and thus would be initially neutral) while subsequent mutation on the pollen component would differentiate the haplotype from the parental one and create a "split" without passing through a self-compatible intermediate. Although the functional model at that time was that of a SR system, they actually produced a S-haplotype that is functionally similar to that of the first stage of scenario "a" from Fig.7 (or purple haplotype in Fig.4). Their verbal scenario of evolution of new compatibilities through dual-specificity stage was criticized at that time by D. Charlesworth and M. Uyenoyama for a number of reasons which would not apply under a CNSR scenario. Hence, it seems to me that the discovery and proposed model of Matton could be discussed in the context of this study.

Overall, I think that this study is of high interest, but the issues raised above should be considered in a manuscript revision.

Minor comments:

-Terminology: the authors refer to different SLF sequences as SLF "alleles". However, I think that calling them alleles is confusing, as some of these sequences correspond to paralogous sequences (the different SLF sequences found in a given S-haplotype; by default 10 at the beginning of the simulation) and some correspond to different allelic copies of a given paralogue in different S-haplotypes (in other words, some sequences are derived vertically and others horizontally by duplication of original SLFs). Hence, I would prefer avoiding the term allele, or at least addressing the issue in the manuscript before using the term.

-Fig4 p 7: the figure is very useful to introduce the notion that compatibility classes are genetically heterogeneous (within class heterogeneity), including for the number of SLF sequences carried. However in the legend it is said that "compatibility classes are defined such as all members of each class are bidirectionally incompatible within the class but simultaneously bidirectionally compatible with all members of all other classes", but this does not hold true for the purple S-RNase haplotype which cannot be fertilized by most of the S-haplotypes of the other two classes (indeed most of those are lacking a SLF with purple recognition). Hence there is incongruity in the figure.

-P.9-10 and fig 6+7: the term "neutral" is used with some confusion. P.9,1224: "the first mutation is neutral", actually it is neutral with respect to the parental haplotype with the original S-RNase (assuming no pollen limitation, see above) but as discussed later it is not necessarily neutral with respect to the other allelic classes (because some may not be able to recognize the new mutant and does the relative fitness of all haplotypes including the new mutant are modified, and thus it is not "fully" neutral).

p.10 fig6: the three scenarios with extinctions of other classes are confusingly referred to in the figure and in the text. In fig 6a they are called extinction (1), (2), (3) while called "cross mut. in non-comp", "neutral", "split+extinction" in fig. 6d and the meaning of "cross mut in non-comp" is not clear before reading Fig.8. In fig.8 they are presented as a b c.

: In the study of Li et al 2017 (Plant Journal) in *Petunia*, the authors suggest that a single amino acid is important for determining the specificity of the SLF. A feature that would not in principle be compatible with promiscuous molecular interactions. This should be discussed.

p.11 l238: "fi is defined as the proportion of diploid individuals in the population it is compatible with as sire". Do the authors take into account the difference in full-compatible and half-compatible (one pollen haplotype only is compatible) pollinations in the calculation of male fitness, which does occur in gametophytic SI?

p.14l313-316f and fig S18: the authors compute the proportion of new S-RNase mutations that are compatible with all non-self classes. This statistic is important because it determines whether the S-RNase mutation is truly neutral or not. However, taking into account the fact that haplotypes within a given allelic class can vary in terms of composition of SLF sequences, how did they compute this statistic? Did they consider a mutation compatible with a given class if at least one of the S-haplotypes of the class was compatible, or if all S-haplotypes of the class were compatible? I suspect that it is the former calculation which has been used, and thus it is overestimating the compatibilities as some proportion of S-haplotypes from presumably "compatible" classes would be incompatible. This should be clarified and discussed.

p.18,l424-426: the authors suggest that the number of SLF sequences in a given haplotype would be a proxy for the number of distinct specificities. However, I think that it is clear from the empirical literature that the number of distinct specificities is well above the number of SLF sequences per haplotype. This crucially depends on the proportion of overlap in recognition of a given SLF sequence, a property that is crucially missing in the results presented in this study (see main comment 3).

Reviewer #2 (Remarks to the Author):

Review of "The role of promiscuous molecular recognition in the evolution of RNase-based self-incompatibility"

I would like to start by saying that I really appreciate this contribution and that although I rarely sign my name to reviews, as the senior author of a highly relevant paper [Harkness et al. *American Naturalist* 2022] - I would like the authors and AE to know that this review is coming from Yaniv Brandvain.

This manuscript does a fantastic job of introducing a plausible mechanistic basis to the mutations underlying SRNASE based SI systems, in a model of the evolution of the gain and loss of novel specificities in this system. This is a difficult and intriguing problem, and incorporating this realistic biology is admirable. I remember early discussion / strategies in developing our own paper in which I knew about promiscuity but could not come up with the appropriate framework to incorporate it, and I think this manuscript is a real advance. Contrary to our manuscript, in which gene conversion played the magic role of providing the variation necessary, the more realistic possibility of mutation and promiscuity is a nice take and I am excited for additional empirical evidence [which will likely come rapidly as high quality de-novo long-read assemblies become more and more affordable] to tease apart the contribution of gene conversion and duplication and mutation of individual residues to the diversification of S-alleles in the SRNASE system.

That is the good. The weaknesses are that the novelty here is on the biophysical mechanism and not necessarily the evolutionary outcome or process. While the path through neutral substitutions is interesting, the dynamics of the model and the ultimate outcomes are much like our paper and Bodega et al. Further, I do not understand what the authors mean when they say that previous "models were unable to determine whether the number of distinct specificities reached an equilibrium." Both Harkness et al and Bodega et al provide equilibrium distributions (see eg figures 8 and 10 of Harkness et al) despite ongoing loss and gain throughout the process (as in this paper). As such, this framing/ justification for novelty should be eliminated, and with this novelty eliminated it is up to the AE to consider whether this real advance is sufficiently important to warrant publication in *Nature Communications*.

-Minor comments-

The default value of proportion of self-pollen on a flower -- alpha -- of 0.95 seems unrealistically

large. The authors do vary this some in the supp, but from what I can tell they do not vary it independently of delta (the extent of inbreeding depression), and it is unclear why. It would be worth seeing how a more realistic value for alpha (perhaps 0.1?) while keeping delta at impacts results.

Additionally the authors vary population size and mutation rate in their sup-which is great. However population size has two distinct effects --it increases the per generation supply mutations by the compound parameter $N\mu$, and impacts the rate of drift. As such, it would be worth investigating parameters in which the authors vary mutation (and duplication and deletion) rates and population size, such that they can distinguish the effect of the compound parameter $N\mu$, from N .

RESPONSE TO REVIEWERS' COMMENTS

Reviewer #1 (Remarks to the Author):

The manuscript by Erez et al. tackles the question of the evolutionary enigma of evolution of new specificities in self-incompatible plants with a collaborative non-self recognition (CNSR) system, such as the S-RNase system of Solanaceae. This theoretical study introduces a very original model that allows for the first time to implement promiscuous molecular recognition between male and female genetic components, a well-known property of CNSR systems which had not been included in theoretical models up to now. It is shown that allowing for promiscuous interactions changes several fundamental outcomes of the models, including evolutionary stability of the number of allelic specificities (which was unclear in previous models), and the introduction of new evolutionary paths that were previously excluded. The implementation of the model and its application are well described overall (see below for some problems of clarity), although it requires very careful attention of the reader to grasp all details. Maybe the authors could find ways to present the model with two levels of description, one simplified presentation highlighting the main results, and one deeper presentation showing all the details.

We thank the reviewer for these warm words.

To simplify the model presentation, we adopted the referee's suggestion and have revised Figs. 7-8 (Figs 6-7 in the revised version). The former figures showed all three split and extinction trajectories and were very detailed. These full figures are now part of the Suppl. File. In the main text, we now show instead only the most common extinction and split trajectories, accompanied by a schematic description of that trajectory.

We have also revised Fig. 3 (now Fig. 2) showing the simulation structure. We believe that the new figures are clearer.

Overall, this study is providing a very important contribution to the understanding of the evolution of CNSR systems. However, beside the question of clarity raised above, the manuscript, in its current version, also does not adequately tackle the following issues, which should be addressed in a revision:

1. Comparison of the observed number of allelic classes as a function of population size with previous theoretical models: here the authors find that the average expected number of allelic classes is about 8-9 for a population size $N=500$. This could have been compared to the classical expectations for Self-Recognition (SR) systems under gametophytic SI produced by Wright and followers. For instance, using Yokoyama and Nei (1978, Genetics) equations, the number of common allelic lines expected with $N=500$ is ranging from $n_a=12,2$ to $n_a=10,3$ for allelic mutation rates varying from 10^{-6} to 10^{-8} . Hence, allele numbers expected under CNSR seem to be lower than under SR. This could be discussed, and could allow some relevant comparisons with empirical results because gametophytic systems with either CNSR (Solanaceae, Plantaginaceae, Maloideae within Rosaceae) or SR (Papaveraceae, Prunus within Rosaceae) have been studied.

We thank the reviewer for this interesting comment. Indeed, calculation of the number of self-incompatibility alleles in a panmictic population assuming SR yielded higher estimates than ours. To compare our values to those produced by the formulas of Wright (1939) and Yokoyama and Nei (1978) we need to first adjust the mutation rate definition, which differs between the models. In the SR models it is assumed that every mutation produces a novel specificity not currently existing in the population and that is also compatible with all existing ones. In our model, the mutation rate is defined per-residue. Most mutations do not produce a novel specificity and even mutations that do so, are not necessarily compatible with all existing population specificities. For a fair comparison, we employ here the birth rate of new classes in the entire population and multiply by the population size, to obtain the per-individual analogue of mutation rate as in previous models. Under our standard parameter values (Fig. 4c), the class birth rate is of the order of 10^{-3} . As the population size is $N=500$, the equivalent per-individual mutation rate is in the range 10^{-5} to 10^{-6} . In our model, we obtained 8 to 9 specificities (compatibility classes) under these parameters. Wright's estimate for $N=500$ and mutation rate $u=10^{-6}$ yielded number of alleles $n_a=12.9$. Yokoyama and Nei's estimates were listed above by the reviewer. We conjecture that the reason for the lower number of specificities in our model is that CNSR needs to keep multiple male-specificity genes to maintain compatibility to all non-self classes. This 'burden' increases, the more classes exist already, which would slow down the emergence of additional classes depending on the number of existing ones. In contrast, for SR the ability to be compatible to all non-self classes is independent of their number, and new specificities only need to be distinguishable from existing ones.

Two more notes are in place: this is indeed a very interesting theoretical question, which warrants an analytical treatment that is beyond the scope of the current work. Following further comments of the reviewer, we discovered that the number of classes increases under pollen limitation (number of fertilization attempts per female, Fig. S23). Pollen limitation was not considered by Wright and Yokoyama & Nei, thereby implicitly assuming pollen abundance.

We added the following text to the discussion (lines 433-442):

"Previous models for the number of self-incompatibility alleles in finite populations assumed SR~\cite{wright_distribution_1939,yokoyama_population_1979,czuppon_revisiting_2022}. These models provide higher estimates of the number of alleles compared to the number of classes obtained in ours (approx. 12 alleles for $N=500$, $\mu=10^{-5}$) compared to 8-9 classes in our case). We hypothesize that this discrepancy is rooted in the different implications on SR vs. NSR of the requirement that every specificity is compatible with all others. A CNSR specificity must maintain multiple compatibilities with all other specificities, a burden that becomes heavier the more specificities exist already. In contrast, an SR specificity only needs to be distinguishable from existing ones, but not to be able to recognize each and every one of them, hence the emergence of novel specificities is much less decelerated with the growing number of existing specificities."

2. Negative frequency-dependent selection acting on the female fitness (Fecundity selection): without explicit recognition, the authors completely obliterate selection acting on female fitness because their implementation, in case of incompatibility between randomly chosen parents "grant multiple opportunities to mate with uniformly chosen non-self pollen grains". Hence, they neglect situations with pollen limitations, although these are often found in nature (Knight et al 2005, *Ann Rev Ecol Evol Syst*). It has been shown that including pollen limitation in theoretical models of SI evolution can greatly impact the evolutionary outcome as it adds an additional

component of negative frequency-dependent selection, coined "fecundity selection" (Vekemans et al. 1998 Evolution). With respect to the model presented here, fecundity selection would potentially greatly impact the evolution of S-RNase mutants (and thus the most frequent scenario in this study where a S-RNase mutation occurs first) because they will suffer from lower (female) fitness if they become incompatible with one or more of the previous allelic classes (in this study, such mutations are assumed to be almost neutral, but this would not be true under pollen limitation models). They recognize that the S-RNase mutant could be non-neutral but in this case they predict a reverse outcome (lower fitness and putative extinction of the other allelic class and not of the S-RNase mutant). It would be important to test the effect of pollen limitation, or at least to discuss this issue.

We thank the reviewer for raising this point. Fecundity selection is captured in our model by the parameter k – the number of fertilization attempts per female (Eq. 6, Methods section). k can be considered as representative of the pollen-to-ovule ratio and can be tuned in the simulation. In our model, if none of the k randomly chosen pollen grains match the maternal plant, this maternal plant will produce no offspring and another maternal plant will be chosen instead. In the results presented in the manuscript, we used $k=5$. For comparison, Harkness et al. (Am Nat 2021) assumed $G=100-1000$ pollen grains and 10 ovules per plant, which is equivalent to $k=10-100$ in our notation and Bodova et al. assumed infinitely abundant pollen ($k=\infty$). To test the effect of pollen limitation in our model we ran the simulation with lower values of $k=1$ and 2, mimicking pollen limitation and also higher values of $k=10, 50$ and 100, mimicking highly abundant pollen. The results are shown in suppl. Figure S23. As the reviewer hypothesized, we found that the ratio between pollen to ovule causes an effective frequency-dependent selection and hence affects the number of classes obtained. For $k=1$ and 2 we observe an increase in the number of classes (13-14 and 10-11 respectively) relative to results shown in the manuscript. For $k=10$ we find a decrease in the number of classes to 6-7. For higher values of $k>10$ the class number is insensitive to k . The intuitive explanation for this dependence on k , is that for lower k there is a higher pressure on the RNase to match a random pollen, which in turn increases the number of classes K , such that the relative proportion of the maternal plant two own classes $2/K$ (whose pollen it rejects) decreases. We also show in Fig. S23c the class death rate for different values of k and find a decrease in that rate for lower k . Indeed, when pollen is abundant there is little pressure on newly formed RNase mutants to match the existing SLF, hence they are very likely to establish in the population and cause extinctions of classes they do not match. In contrast, if pollen is limited, new RNases incompatible with the existing population SLFs are less likely to establish and hence fewer class extinctions occur.

We have also modified the text defining the male-fitness, to better clarify our approach (line 235-240):

"...their **male-fitness** f_i , and 'fitness-adjusted copy numbers' $\tilde{X}_i = X_i + (f_i - \bar{f}) \cdot N$ (Methods) against time, where \bar{f} is the population mean **male-fitness**. Haplotype **male-fitness** f_i is defined as the proportion of diploid individuals in the population the haplotype is compatible with as sire. While reproductive capacity is affected by both male and female components, we find that unless pollen limitation is imposed, the haplotype dynamics is well-captured by the male-fitness alone. We illustrate the effect of pollen limitation in Fig. S23."

We also refer to increase in class number due to pollen limitation in the discussion (lines 426-428):

“The number of compatibility classes in our model positively correlates with the population size and Eth... and also increases under extreme pollen limitation.”

3. Expected level of molecular promiscuous interactions: the originality of the model is that it allows promiscuous interactions, i.e. a single SLF molecule can recognize (and detoxify) several S-RNase molecules, including some that are not (yet) present in the population. This property is the basis of the original results found in the outcome of the model, as compared to previous models. However, it is frustrating that the authors do not present results on the distribution of this overlap of functional recognition. This information seems to me to be crucial to understand properties of the model, although obviously it will depend on model parameters. It is also very interesting because it could be compared with empirical estimates. Such estimates are still rare in the literature but many groups are currently investigating molecular approaches to tackle this issue empirically (with functional tests). Hence, producing expectations for the average number S-RNase alleles recognized by a single SLF molecule, and for the variance in overlap between different SLF molecules, would be very useful. I don't understand why this has not been addressed by the authors.

We have added this information in an additional supplementary figure, Fig. S21. Fig. S21a shows the distribution of the effective number of distinct RNases that a single SLF can detoxify. The average number of RNases recognized by a single SLF is nearly 2, but the distribution is highly skewed. Interestingly we observe that a significant proportion (>25%) of the SLFs are dysfunctional, namely they do not detoxify any of the population RNases.

Complementary to that we show in Fig. S21b the recognition overlap between same-haplotype unique SLFs. In S21c we show the distribution of the number of shared RNases between all possible pairs of same-haplotype SLFs. These two panels demonstrate the redundancies amongst same-haplotype SLFs. Given these redundancies, we illustrate in Fig. S21d the distribution of the minimal sufficient number of SLFs. Namely, the minimal number of SLFs that could recognize all non-self classes. These distributions peak at 4 or 5 (depending on the number of population classes). For comparison, the mean number of SLFs per haplotype is 10 (Fig. S20), demonstrating a widespread redundancy.

We refer to these new figures in the discussion, where we mention the interaction promiscuity.

We agree with the reviewer that this is a very interesting aspect of the model and hence we believe it warrants a deeper investigation. We have an additional manuscript currently in preparation, in which we will explore this in more depth.

4. Variation in the number of SLF sequences per S-haplotype: in the model, diversity in the interaction properties of SLF sequences can arise because of duplication and mutation or because of mutation only. The variability in the number of SLF sequences per S-haplotype is of interest, and again especially because it could be compared with empirical observations. It would be interesting to have expected results about this variability, and to discuss those.

We have added an additional supplementary figure, Fig. S20 showing the distribution of the number of SLF sequences per haplotype. We note here that we also ran the simulation with no gene duplication or deletion, such that the number of SLF sequences per haplotype remains

constant (9 with the standard parameters) and found that the effect on the class number distribution is negligible (Fig. S13). We believe that this is because a significant proportion of SLF sequences are dysfunctional, hence small variations in their number does not affect the model macroscopic behavior.

5. Also of interest, would be to compute the level of "completeness" (the proportion of S-haplotypes that can detoxify, based on their set of SLF sequences, all members of all other classes) of S-haplotypes in the population, as this has been shown to be an important variable in previous models.

We note here that the level of completeness is closely related to the proportion of classified haplotypes, that was shown already in a few examples. Every haplotype that is affiliated with a class is by definition bidirectionally compatible with all members of all classes except its own class and hence is complete. The proportion of 'complete' haplotypes could slightly exceed the proportion of classified ones, if there are complete haplotypes that are also self-compatible or haplotypes that became compatible with some of their class members and hence were removed from their class. Note that the proportion of classified haplotypes is rather high: 96% on average with our default parameter values, and between 90%-99% for other choices of parameter values we checked (Figs. S3-S12). For completeness of presentation, we have added the proportion of complete haplotypes under our baseline parameters – Fig. S27. Under these parameter values 0.96 of the haplotypes are classified and a slightly larger fraction of 0.967 of the haplotypes are complete. Only the remainder of 0.033 fraction of haplotypes is incomplete.

6. Full inbreeding depression: in most simulations, the inbreeding coefficient is set to 100%. Hence, under these circumstances all scenarios with a self-compatible intermediate are completely disfavored (see figure 6C). I think this should be more explicitly discussed, as this artificially increases the contrast in outcome of this study as compared to previous studies (using full inbreeding depression will favor scenarios that do not lead to self-compatible intermediates).

We thank the reviewer for this comment. We have also analyzed the frequencies of the different diversification trajectories for weaker inbreeding depression values – see Fig. S17. Indeed, the frequencies of the three additional trajectories going through SC intermediates increases under lower inbreeding depression, as the reviewer correctly predicted, yet remain secondary to the trajectories maintaining SI throughout.

Decreasing δ and α further resulted in the takeover of the population by SC variants, rather than by different diversification trajectories going through SC. For $\delta = 0.8$ either 100% of the runs (for $\alpha=0.6$) or 76% of the runs (for $\alpha = 0.8$) ended when the population was taken over by SC (Figs. S24-S26).

We also note that the requirement for high inbreeding depression to maintain self-incompatibility is in agreement with previous models. Bodova et al. (Genetics, 2018) comprehensively mapped the α - δ parameter range (Fig. 5, there) and found that unless $\delta > 0.85$, the population is dominated by SC genotypes.

Harkness et al. (2021) assumed $\delta=1$ throughout “We model extreme inbreeding depression such that all selfed offspring are inviable, so the selfing rate is equivalent to a fecundity penalty.”

Under the inbreeding depression conditions for which SI genotypes still dominate the population the trajectories going through SC remain the least common (for example, for $\delta=0.9$, $\alpha=0.6$ these 3 trajectories together sum up to 16% of the diversification events documented, Fig. S17).

We have modified the text accordingly to discuss this (lines 222-227):

“In principle, the three mutations needed for a split could occur in six different orderings, but the other three options pass through a self-compatible intermediate and hence are **much less common under the high inbreeding depression needed to maintain self-incompatibility**. In \figref{fig:split_extinction_scheme} we show the simulation frequencies of the six optional split trajectories. These could potentially vary depending on the simulation parameter values **and specifically inbreeding depression** (see Figs. S15-S17)”

We are also discussing this point in the Discussion (lines 395-400):

“Similar to previous models \cite{bodova_evolutionary_2018,harkness_diversification_2020}, maintenance of self-incompatibility requires high inbreeding depression (Figs. S25-27). Inbreeding depression affects the relative proportions of the different diversification trajectories, and specifically under lower inbreeding depression, trajectories passing through a self-compatible intermediate become more common, yet remain second to the three trajectories maintaining SI throughout.”

And lines 349-350:

“Trajectories traversing through a self-compatible intermediate are possible too in our model, but are much less common (Fig. 5c, Fig. S17).”

7. Reference to the literature on dual-specificity S-RNase mutants: Matton et al 1999 (Plant Cell) found in *Solanum chacoense* that by experimentally altering amino acids of an S-RNase sequence, they generated a so-called "dual-specificity" mutant that behaves initially similarly to its parental allele (and thus would be initially neutral) while subsequent mutation on the pollen component would differentiate the haplotype from the parental one and create a "split" without passing through a self-compatible intermediate. Although the functional model at that time was that of a SR system, they actually produced a S-haplotype that is functionally similar to that of the first stage of scenario "a" from Fig.7 (or purple haplotype in Fig.4). Their verbal scenario of evolution of new compatibilities through dual-specificity stage was criticized at that time by D. Charlesworth and M. Uyenoyama for a number of reasons which would not apply under a CNSR scenario. Hence, it seems to me that the discovery and proposed model of Matton could be discussed in the context of this study.

We thank the reviewer for highlighting the relation of this empirical result to our model. We have added a sentence in the Discussion referring to Matton et al. 1999:

“Such neutral variation amongst RNases was indeed detected in natural *\textit{Pyrus communis}* populations ~\cite{sanzol_two_2010} and a putative ancestral mutant proposed to precede a split was demonstrated in *\textit{Solanum chacoense}* \cite{matton_production_1999}. “

We hope that our model will stimulate further empirical demonstration of such RNase mutations.

Overall, I think that this study is of high interest, but the issues raised above should be considered in a manuscript revision.

We thank the reviewer for the positive evaluation of our manuscript, and for the thorough reading and the many insightful comments that helped us improve it and sharpen several points.

Minor comments:

8. -Terminology: the authors refer to different SLF sequences as SLF "alleles". However, I think that calling them alleles is confusing, as some of these sequences correspond to paralogous sequences (the different SLF sequences found in a given S-haplotype; by default 10 at the beginning of the simulation) and some correspond to different allelic copies of a given paralogue in different S-haplotypes (in other words, some sequences are derived vertically and others horizontally by duplication of original SLFs). Hence, I would prefer avoiding the term allele, or at least addressing the issue in the manuscript before using the term.

We thank the reviewer for highlighting this ambiguity. Indeed, we were using the term 'alleles' in multiple contexts, as the reviewer noted. In addition, we also used the term 'alleles' to refer to the number of SI specificities, formerly also called 'S-alleles' – following previous studies that focused on SR systems for which every allele constituted a different specificity.

We have changed the terminology, and now wherever we refer to same haplotype SLFs (SLFs derived horizontally) we now call them 'SLF paralogs'. Wherever 'alleles' referred 'SI specificities' (S-alleles) we changed this into 'specificities'. 'sequence' is used when we mean 'any gene'.

The word 'allele' is still used in reference to different RNases variants. RNases are not duplicated or deleted in our model, hence they are only derived vertically.

Below are some examples for text changes (we changed throughout the manuscript).

Abstract:

"...previous models struggled to pinpoint the evolutionary trajectories by which new **specificities** evolved. Here, we construct a novel theoretical framework, that crucially affords interaction promiscuity and multiple distinct partners per protein, empirical findings disregarded by previous models. "

Line 415:

"The number of **SLF paralogs** per haplotype determined empirically..."

Lines 434-436:

"Previous models for the number of **self-incompatibility specificities** in finite populations assumed SR. These models provide higher estimates of the number of **specificities** compared to the number of classes obtained in ours"

Table 1 (Methods):

" $n_h=9$ & initial number of **SLF paralogs** per haplotype"

Lines 566-569:

"**Gene sequence** indices are set uniquely, such that every new variant receives a unique index, not given previously to any other one, including sequences that no longer exist in the population.

If the same sequence emerged independently more than once, it received the same index upon each emergence.”

9. -Fig4 p 7: the figure is very useful to introduce the notion that compatibility classes are genetically heterogeneous (within class heterogeneity), including for the number of SLF sequences carried. However in the legend it is said that "compatibility classes are defined such as all members of each class are bidirectionally incompatible within the class but simultaneously bidirectionally compatible with all members of all other classes", but this does not hold true for the purple S-RNase haplotype which cannot be fertilized by most of the S-haplotypes of the other two classes (indeed most of those are lacking a SLF with purple recognition). Hence there is incongruity in the figure.

We thank the referee for noticing this glitch. We corrected the colors in the figure (now Fig. 3), as pointed out by the referee.

10. -P.9-10 and fig 6+7: the term "neutral" is used with some confusion. P.9,1224: "the first mutation is neutral", actually it is neutral with respect to the parental haplotype with the original S-RNase (assuming no pollen limitation, see above) but as discussed later it is not necessarily neutral with respect to the other allelic classes (because some may not be able to recognize the new mutant and does the relative fitness of all haplotypes including the new mutant are modified, and thus it is not "fully" neutral).

We thank the reviewer for highlighting this point. Compatibility of an RNase mutation with a class is not a binary property, because the RNase could be compatible with a subset of the class haplotypes and incompatible with the rest (see also our answer to comment #14 below and Fig. S19), since the classes can be genetically heterogeneous. Two final outcomes are then possible in cases of partial compatibility. Either the class changes its composition, gains full compatibility with the new RNase and survives or it is unable to restore full compatibility and thus becomes extinct. We emphasize, that a class is not necessarily extinct if some of its haplotypes are incompatible but on the other hand, it does not necessarily survive even if a subset of its haplotypes maintains compatibility. Likely, the final outcome is affected by the fraction of the class that is compatible, which we quantify in Fig. S19. In the case of partial compatibility, the fitness of the subset of haplotypes affected is modified, and hence the RNase mutation is not necessarily neutral with respect to the entire population, as the reviewer correctly noted.

Our presentation in Fig. 5b (formerly 6b) of neutral vs. non-neutral RNase mutations was meant to convey the main message that the occurrence of an RNase mutation could end up with all classes fully compatible with the mutation or with the extinction of one or more classes. We have changed the terminology in the figure to be 'essentially (non)-neutral' instead.

We have modified the text in line 220 to be: "In all trajectories the first mutation is **essentially** neutral..." and added a footnote clarifying this point:

"The first and second trajectories begin with an RNase mutation that does not cause the extinction of any of the existing classes, hence we consider it to be 'essentially neutral'. Yet, such an RNase mutation could be incompatible with a subset of one (or more) of the classes (that the ancestral RNase was compatible with!) and hence lead to change of composition of that class, though not to its extinction. In that case, the fitness of the subset of haplotypes affected is

modified, and hence the RNase mutation is not strictly neutral with respect to the entire population. For brevity, we refer below to 'neutral' in this meaning.”

We modified the caption of Fig. 5b as follows:

“An essentially neutral RNase mutation (top) means that all foreign classes are either fully compatible as sires with the new RNase (red) similar to the original RNase (blue) or partially compatible but can adjust by changing their composition to regain full compatibility. An essentially non-neutral RNase mutation (bottom) means that at least one non-self class is (at least partially) sire-incompatible with the mutant RNase to the extent that it fails to adjust and restore full compatibility. For brevity we omit below the word 'essentially' and use the term neutral in this sense.”

The text in figures 6-7 (formerly 7-8) was changed and they only refer to 'original RNase' and 'mutant RNase' and do not mention RNase neutrality.

11. p.10 fig6: the three scenarios with extinctions of other classes are confusingly referred to in the figure and in the text. In fig 6a they are called extinction (1), (2), (3) while called "cross mut. in non-comp", "neutral", "split+extinction" in fig. 6d and the meaning of "cross mut in non-comp" is not clear before reading Fig.8. In fig.8 they are presented as a b c.

We thank the referee for noticing this inconsistency. We changed the text in panel d (now this is Fig. 5) into extinction 1, 2, 3 to conform to panel a.

12. In the study of Li et al 2017 (Plant Journal) in Petunia, the authors suggest that a single amino acid is important for determining the specificity of the SLF. A feature that would not in principle be compatible with promiscuous molecular interactions. This should be discussed.

Li et al studied two functionally distinct SLFs in Petunia that differ in position 293, which is also under strong positive selection: PhS3-SLF1 has histidine (H) and PhS3L-SLF1 has glutamic acid (E). They produced transgenes in which they swapped site 293, namely produced: PhS3-SLF1-H293E and PhS3L-SLF1-E293H. They calculated the electrostatic potentials of these mutants and tested their effect on the plant self-incompatibility. They demonstrate both a change in electrostatic potential (though asymmetric between the mutations) and a change in the compatibility phenotypes following this single amino acid substitutions. Yet, these results refer only to 2 SLFs, and the authors clearly state that a larger number of amino acids are involved in specificity determination, distinguishing these SLFs from others, not included in that study.

Molecular recognition in our model is determined by a combination of L amino acids, but the model does not specify the number of amino acids distinguishing pairs of functionally different RNases or SLFs. In particular, it is possible in our model that two functionally distinct SLFs differ in only one position. For example, in the split trajectories, we demonstrate cases in which a single mutation changed the specificity of RNase or SLF and facilitated the formation of a new class. How the sequences of different classes organize in the sequence space to be functionally distinguishable but simultaneously maintain interactions with many partners is a topic of much interest, which will be covered in more depth in our next publication (currently in preparation).

We have added a mention of this empirical result in the discussion (lines 407-408): “These include estimation of the number of residues distinguishing between functionally different RNase alleles~\cite{matton_hypervariable_1997} and functionally different SLFs~\cite{li_electrostatic_2017},”

13. p.11|238: “ f_i is defined as the proportion of diploid individuals in the population it is compatible with as sire”. Do the authors take into account the difference in full-compatible and half-compatible (one pollen haplotype only is compatible) pollinations in the calculation of male fitness, which does occur in gametophytic SI?

The sentence cited here is ambiguous and we apologize for this. By “it” we meant a haplotype, and not the maternal plant, as could have been implied.

We rephrased the sentence (lines 236-238) which now reads:

“Haplotype male-fitness f_i is defined as the proportion of diploid individuals in the population **the haplotype** is compatible with as sire.”

14. p.14|313-316f and fig S18: the authors compute the proportion of new S-RNase mutations that are compatible with all non-self classes. This statistic is important because it determines whether the S-RNase mutation is truly neutral or not. However, taking into account the fact that haplotypes within a given allelic class can vary in terms of composition of SLF sequences, how did they compute this statistic? Did they consider a mutation compatible with a given class if at least one of the S-haplotypes of the class was compatible, or if all S-haplotypes of the class were compatible? I suspect that it is the former calculation which has been used, and thus it is overestimating the compatibilities as some proportion of S-haplotypes from presumably “compatible” classes would be incompatible. This should be clarified and discussed.

We thank the reviewer for this comment. See also our answer to comment #10 above. Indeed, the property of a haplotype compatibility with a class is not binary, because it can be compatible with different fractions of the class, as the reviewer correctly noted. In the former version of Fig. S18 we used a threshold of 25% (which was not mentioned by mistake), namely a class was compatible with a mutation if at least 25% of the class haplotypes were compatible with it. As classes are heterogeneous and can adjust their composition upon the occurrence of an RNase mutation in another class, a 100% compatibility is not necessary for the class to survive. On the other hand, a class could become extinct even if some fraction of it is incompatible with the mutation, and 0% compatibility is not a pre-requisite for extinction. We revised that figure as Fig. S19. We are currently showing the number of classes incompatible with RNase mutations for 2 threshold values: 100% and 75% compatibility of the class with the mutation. We have added an additional panel showing the distribution of the class fraction compatible with the RNase mutation for all cases with less than 100% compatibility.

15. p.18,|424-426: the authors suggest that the number of SLF sequences in a given haplotype would be a proxy for the number of distinct specificities. However, I think that it is clear from the empirical literature that the number of distinct specificities is well above the number of SLF sequences per haplotype. This crucially depends on the proportion of overlap in recognition of a given SLF sequence, a property that is crucially missing in the results presented in this study (see main comment 3).

We thank the reviewer for this comment. Indeed the former statement was inaccurate, as the reviewer correctly noted. The number of population specificities also depends on the number of RNases detoxified by each SLF and on the number of SLF paralogs that recognize a single RNase. If we describe these RNase-SLF interactions as a bipartite graph where SLF paralogs are nodes in one side of the graph and all the population RNases are nodes in the other side, then these are the SLF and RNase node degrees. We have added this information in the revised version as Figs. S21.

We have also modified the text in the discussion accordingly. We note also that the number of specificities, namely compatibility classes, (8-9 under baseline parameters) in our model is slightly smaller than the number of SLF sequences per haplotype (on average 10), whereas in nature it is the other way around. The reason for this discrepancy could be spatial heterogeneity in natural populations, such that SLFs evolve to only recognize RNase specificities in their neighborhood, but not necessarily specificities found further away, which could have been counted in such surveys.

corrected text (lines 415-425):

“The number of SLF genes per haplotype determined empirically for species with CNSR SI were 16-20 for *\textit{Pyrus}*~\cite{claessen_finding_2019}, *\textit{Malus}*~\cite{pratas_inferences_2018} and *\textit{Petunia}*~\cite{wu_sequence_2020,williams_transcriptome_2014}, but 32 for *\textit{Antirrhinum}* \cite{zhu_snapdragon_2023}. The number of distinct RNases found in natural population surveys were 20 for *\textit{Sorbus aucuparia}*~\cite{raspe_population_2007} and 25 for *\textit{Pyrus pyraaster}*~\cite{hoebee_diversity_2012}, but these may be under-estimates due to limited sample sizes. The number of SI is specificities in a species is thought to exceed the number of per-haplotype SLF genes: for example in *\textit{Malus}* 34 specificities were identified, but only 17-19 SLFs per haplotype~\cite{pratas_inferences_2018}.

The number of compatibility classes in our model (8-9 under baseline parameters) is slightly smaller than the number of SLF sequences per haplotype (on average 10 - \figref{figSI:SLFs_per_haplotype}) though. A possible reason for this discrepancy could be spatial heterogeneity in natural populations, such that SLFs evolve to only recognize RNase specificities in their vicinity, but not necessarily with more distant ones, whereas our model assumes a panmictic population.”

Reviewer #2 (Remarks to the Author):

Review of "The role of promiscuous molecular recognition in the evolution of RNase-based self-incompatibility"

I would like to start by saying that really appreciate this contribution and that although I rarely sign my name to reviews, as the senior author of a highly relevant paper [Harkness et al. *American Naturalist* 2022] - I would like the authors and AE to know that this review is coming from Yaniv Brandvain.

This manuscript does a fantastic job of introducing a plausible mechanistic basis to the mutations underlying SRNASE based SI systems, in a model of the evolution of the gain and loss of novel specificities in this system. This is a difficult and intriguing problem, and incorporating this realistic biology is admirable. I remember early discussion / strategies in developing our own paper in which knew about promiscuity but could not come up with the appropriate framework to incorporate it, and intros way this manuscript is a real advance. Contrary to our manuscript, in which. Gene conversion played the magic role of providing the variation necessary, the more realistic possibility of mutation and promiscuity is a nice take-and I am excited for additional empirical evidence [which will likely come rapidly as high quality de-novo long-read assemblies become more and more affordable] to tease apart the contribution of gene conversion and duplication and mutation of individual residues to the diversification of S-alleles in the SRNASE system.

We thank the reviewer for the positive evaluation of the contribution of our manuscript.

16. That is the good. The weaknesses are that the novelty here is on the biophysical mechanism and not necessarily the evolutionary outcome or process. While the path through neutral substitutions interesting, the dynamics of the model and the ultimate outcomes are much like our paper and Bodega et al. Further, I do not understand what the authors mean when they say that previous "models were unable to determine whether the number of distinct specificities reached an equilibrium." Both Harkness et al and Bodova et al provide equilibrium distributions (see eg figures 8 and 10 of Harkness et al) despite ongoing loss and gain throughout the process (as in this paper). As such, this framing/ justification for novelty should be eliminated, and with this novelty eliminated it is up to the AE to consider whether this real advance is sufficiently important to warrant publication in *Nature Communications*.

While a stable equilibrium in the number of classes is a property of our model, our model is not the first one to exhibit equilibrium, as the reviewer wrote, and we apologize for this misunderstanding. We have removed these claims (including the sentence mentioned by the reviewer) from the text and rephrased the abstract and discussion accordingly, as the reviewer can appreciate.

The novelty of our model goes beyond the introduction of our unique molecular recognition model that allows for multiple partners per protein. This biophysical model opens up new avenues that could not have been reached by the previous models, as we detail below.

Building on this biophysical model, we show that diversification can occur under more permissive conditions than previously thought. Thanks to the promiscuity property, which is unique to our model, diversification does not rely on the occurrence of either gene conversion

(as in Harkness et al.) or on repeated mutations on every genetic background (as in Bodova et al.) to restore compatibility of a new mutant RNase with the existing classes. Instead, it only needs to gain compatibility with its class of origin. As such, our model works under a broad range of parameter values.

Some of our evolutionary diversification trajectories are closely related to those previously pointed out by Bodova et al. However, Bodova et al. found that the more common trajectories traverse through an SC intermediate, which is not needed in the new trajectories we find. The introduction of the biophysical model opens up the new possibility of essentially neutral RNase mutation, which are a prerequisite for the two most common trajectories we show. Previous models were unable to describe neutral RNase mutations, and hence overlooked these trajectories. Empirical evidence suggests that neutral variation and promiscuity of interactions indeed exist (e.g. Matton et al, Sanzol et al., Zhao et al.) but their theoretical implication have not been considered before.

Harkness et al. pursued a different path for diversification that relied on gene conversion – a process not included in our model, nor in the model of Bodova et al. Likely, both gene conversion and mutations-only diversifications occur in nature and the relative contribution of either option is a question to be settled empirically.

Lastly, our model also demonstrates for the first time the natural organization of the population into compatibility classes. In models assuming simplified discrete allele description, the existence of such classes is pre-assumed and is not shown to naturally evolve. In contrast, our model affords a more complicated description, where classes in which almost all genotypes are ‘complete’ is not the only option. Yet, we show that almost all genotype naturally self-organize into such classes and this happens under a broad range of the model parameters.

-Minor comments-

17. The default value of proportion of self-pollen on a flower -- α -- of 0.95 seems unrealistically large. The authors do vary this some in the supp, but from what I can tell they do not vary it independently of δ (the extent of inbreeding depression), and it is unclear why. It would be worth seeing how a more realistic value for α (perhaps 0.1?) while keeping δ at impacts results.

First, we clarify that we define α is the proportion of self-pollen reaching the flower out of the total pollen received, rather than the proportion out of the total pollen of the plant that remains on its own flowers, following the definition in Bodova et al. 2018.

Following this comment, we ran additional simulation of the model with various values of α . In Figs. S24 and S25 we keep δ constant and only vary α . In Fig. S24 we show simulation results with $\delta=0.9$ and $\alpha=0.1, 0.4, 0.5, 0.6$ and 0.8 . We noticed that for low α values the population is more likely to be taken over by self-compatible genotypes. We plot there the population SC proportion against time and stop the run if the SC fraction exceeds 80% of the entire population. For $\alpha=0.6$ and 0.8 none of the runs reached the 80% threshold, but for $\alpha=0.5$ 68% of the runs and for $\alpha=0.1$ and 0.4 100% of the runs reached the SC 80% limit within our run time. For a lower value of δ , a higher value of α

α is needed to avoid takeover by SC genotypes. Indeed, with $\delta=0.8$, we found that for $\alpha=0.6$, 100% of the runs reached SC, and for $\alpha=0.8$, 76% of them did. We summarize these results in Fig. S26.

We note that this requirement for high enough values of α to maintain self-incompatibility are in agreement with the results of Bodova et al (2018). Fig. 5A there shows a thorough mapping of the population proportion of SI haplotypes for different combinations of α and δ values with good agreement with our numbers: <https://academic.oup.com/genetics/article/209/3/861/5930990>

Measurement of α is complicated and thus scarce and could also vary between species and depend on other conditions, such as the ubiquity of pollinators and the population density. Such a measurement was reported for white clover (*Trifolium repens*) and estimated the fraction of pollen originating from the very same flower to be >50% of the total pollen received. As an unknown fraction of the remaining pollen brought by pollinators comes from different flowers of the same plant (that is also self-pollen) the authors conclude that the overall proportion of self pollen is even higher than 50%.

See Rodet et al., "Status of self-pollen in bee pollination efficiency of white clover (*Trifolium repens* L.)", *Oecologia*, 1998.

18. Additionally the authors vary population size and mutation rate in their sup-which is great. However population size has two distinct effects --it increases the per generation supply mutations by the compound parameter $N\mu$, and impacts the rate of drift. As such, it would be worth investigating parameters in which the authors vary mutation (and duplication and deletion) rates and population size, such that they can distinguish the effect of the compound parameter $N\mu$, from N .

We thank the reviewer for this interesting comment. Indeed, the mean number of mutations per generation in the entire population (the mutation flux) is $N\mu$, as the reviewer wrote. Thus, for a large number of classes and if their sizes are equal and large enough, this compound parameter is the main determinant for the appearance of the initiating mutations in our split and extinction trajectories. It is thus natural to test the relation of $N\mu$ to the class number and birth and death rates. However, both split and extinction are multi-stage processes that require a series of several mutations of different kinds, with additional restrictions. For example, the probability that an RNase mutation will match one of the SLFs in its own class, required for split trajectories, does not seem to depend on $N\mu$ but surely affects the likelihood of split completion. Because of this complicated dependence of splits on the supply of mutations with specific properties, we do not expect $N\mu$ alone to fully explain the number of classes or their birth and death rates. We ran additional simulations with different combinations of N and μ values such that their product $N\mu$ remains constant – see Fig. S22. From the results we learn that $N\mu$ can explain the number of classes for large, but not for small population size. We conclude that the probabilities of some of the mutations in the process have different functional dependence on N and μ . Further investigation of the exact mathematical dependence of the number of classes on the model parameters is beyond the scope of the current manuscript and we leave it for future work.

REVIEWERS' COMMENTS

Reviewer #1 (Remarks to the Author):

I read carefully the response to my previous review (as reviewer 1) of the manuscript by Erez et al., and read the revised version. I think the authors understood my many comments and did an excellent work of responding point by point, making relevant changes in the text and even providing additional results (for the fecundity selection issue). Hence, I'm highly satisfied with the revision and do not have additional comments, except perhaps that the authors could have cited previous work on the effect of fecundity selection on the number of specificities maintained in populations, as currently they seem to reinvent the wheel in this work.

Prof. Xavier Vekemans, University of Lille, France.

Reviewer #2 (Remarks to the Author):

This is a nice contribution to the literature and should be published one data and code are made available

Reviewer #2 (Remarks on code availability):

There is no repo at this link, nor is there any such repo in the Tamar Friedlander Lab github. As such the code could not be reviewed.

RESPONSE TO REVIEWERS' COMMENTS

The reviewer comments are in regular black font and our response is in blue

.

Reviewer #1 (Remarks to the Author):

I read carefully the response to my previous review (as reviewer 1) of the manuscript by Erez et al., and read the revised version. I think the authors understood my many comments and did an excellent work of responding point by point, making relevant changes in the text and even providing additional results (for the fecundity selection issue). Hence, I'm highly satisfied with the revision and do not have additional comments, except perhaps that the authors could have cited previous work on the effect of fecundity selection on the number of specificities maintained in populations, as currently they seem to reinvent the wheel in this work.

Prof. Xavier Vekemans, University of Lille, France.

We thank the reviewer for these encouraging words and for many insightful comments that helped us improve our manuscript.

We have added as additional reference the paper "Mate availability and fecundity selection..." By Vekemans et al., *Evolution* 1998.

We apologize that this comment slipped our attention in the previous revision.

Reviewer #2 (Remarks to the Author):

This is a nice contribution to the literature and should be published one data and code are made available.

Reviewer #2 (Remarks on code availability):

There is no repo at this link, nor is there any such repo in the Tamar Friedlander Lab github. As such the code could not be reviewed.

We thank the reviewer for the positive evaluation of our work.

We have made the code publicly available in the following link:

https://github.com/Tamar-Friedlander-Lab/evol-RNase-based-SI_stochast_simul